# POLICY REHEARSING: TRAINING GENERALIZABLE POLICIES FOR REINFORCEMENT LEARNING

**Chengxing Jia**[1,2][*] **Chen-Xiao Gao**[1][*] **Hao Yin**[1]**, Fuxiang Zhang**[1,2]**, Xiong-Hui Chen**[1,2]**,**
**Tian Xu**[1,2]**, Lei Yuan**[1,2]**, Zongzhang Zhang**[1]**, Zhi-Hua Zhou**[1]**, Yang Yu**[1,2][†]
[1]National Key Laboratory for Novel Software Technology, Nanjing University, China &
School of Artificial Intelligence, Nanjing University, China
[2]Polixir Technologies
`{jiacx,gaocx,yinh,zhangfx,chenxh,xut,yuanl}@lamda.nju.edu.cn`
`{zzzhang,zhouzh,yuy}@nju.edu.cn`

## ABSTRACT

Human beings can make adaptive decisions in a preparatory manner, i.e., by making preparations in advance, which offers significant advantages in scenarios where both online and offline experiences are expensive and limited. Meanwhile, current reinforcement learning methods commonly rely on numerous environment interactions but hardly obtain generalizable policies. In this paper, we introduce the idea of *rehearsal* into policy optimization, where the agent plans for all possible outcomes in mind and acts adaptively according to actual responses from the environment. To effectively rehearse, we propose ReDM, an algorithm that generates a diverse and eligible set of dynamics models and then rehearse the policy via adaptive training on the generated model set. Rehearsal enables the policy to make decision plans for various hypothetical dynamics and to naturally generalize to previously unseen environments. Our experimental results demonstrate that ReDM is capable of learning a valid policy solely through rehearsal, even with *zero* interaction data. We further extend ReDM to scenarios where limited or mismatched interaction data is available, and our experimental results reveal that ReDM produces high-performing policies compared to other offline RL baselines.

## 1 INTRODUCTION

In Reinforcement Learning (RL), policies are typically optimized through intensive interactions either with the target environment (Sutton & Barto, 2018) or with simulators as a proxy environment (Tobin et al., 2017; Rao et al., 2020). However, it is challenging for real-world applications since online real-world interactions are costly and sometimes dangerous, and building high-fidelity simulators requires a huge amount of labor work and expertise. Offline RL (Levine et al., 2020) optimizes the policy with pre-collected datasets rather than online interactions (Fujimoto et al., 2019; Kumar et al., 2019; 2020; Kostrikov et al., 2022) and has made remarkable progress recently (Prudencio et al., 2022). However, obtaining a comprehensive dataset, which is essential for developing a reliable offline policy, can be costly in real-world applications (Kiran et al., 2021). This leads to the question: Can the agent effectively make decisions without online interaction or abundant offline data?

To adapt to unfamiliar or intricate situations, humans employ a cognitive mechanism known as rehearsal (Cowan & Vergauwe, 2015; Ignacio et al., 2016; Oberauer, 2019) that involves mentally practicing actions and envisioning potential outcomes. When all conceivable outcomes have been rehearsed in minds, human could make adaptive decisions based on real-world situations (Zhou, 2022). Prior works have involved the concept of rehearsal in machine learning, such as continual learning (Yoon et al., 2022), causality (Churamani et al., 2023) and neuro-symbolic (Marconato et al., 2023). In this work, we consider whether a policy, by incorporating rehearsal similar to humans before taking actions, can effectively make decisions under limited real-world data.

---

[*]These authors contributed equally. Work was done during an internship at Polixir Technologies.
[†]Yang Yu is the corresponding author.

For the mentioned consideration, we incorporate rehearsal into the reinforcement learning process to better emulate human-like learning, which we refer to as *policy rehearsing*, and propose Policy **Re**hearsing via **D**ynamics **M**odel Generation (ReDM) to develop a policy that can achieve satisfactory performance in environments without interaction or rich offline data. The framework of policy rehearsing iterates between two procedures. The first is to generate various tasks. Similar to human rehearsal, these tasks may not directly reflect the real environment but each model represents a unique and plausible hypothesis about it. Since reward functions are often designed according to the task goals, we mainly focus on the hypothesis on dynamics model space. The second trains a meta-policy that can adapt to generated dynamics models. Through the ongoing generation of new dynamics models and simultaneous meta-policy learning, the policy could make rehearsals for more situations and its ability to adapt to the potential target environment gradually strengthens, despite not interacting with the target environment. However, implementing the mechanism of policy rehearsing is challenging, as the whole hypothesis space of dynamics model is large. Therefore, the core of policy rehearsing lies in how to reduce the hypothesis space of candidate dynamics models. To achieve this goal, we propose that dynamics model generation should adhere to the principles of *diversity* and *eligibility*. More specifically, we enforce diversity among the candidates by iteratively generating new candidates to minimize the worst-case performance of the current policy. While eligibility, which measures the validity of dynamics, is optimized by constraining the model to produce viable paths towards high returns. By incorporating diversity and eligibility, we effectively shrink the hypothesis space for generating candidate dynamics models and subsequently employ the meta-policy for learning.

As our model generation process is not based on directly mimicking the target, it does not necessarily require interactive data. Instead, it relies on some task knowledge that can be readily obtained in practice, such as the reward function and the terminal function. Our experiments on low-dimension control tasks demonstrate that even with NO interaction data, ReDM is capable of producing a valid policy that significantly outperforms the random policy. In order to extend its applicability to more complex tasks, where hypothesis space is far too large, the framework of ReDM is also compatible with offline datasets or demonstrations by using the data as a regularizer for generating models. However, it is important to note that the interaction data is only used to narrow down the hypothesis space, and thus even if the provided data is limited in quantity and coverage or slightly inconsistent in dynamics, the adaptive policy produced by ReDM remains competitive. This sets ReDM apart from traditional offline RL methods where the unbiasedness of data is of vital importance for policy optimization. To validate this, we test ReDM with limited or misspecified data from D4RL, a widely used benchmark for offline RL, and it turns out that ReDM outperforms other baseline methods including state-of-the-art model-free and model-based offline RL algorithms. The results support that policy rehearsing is a effective approach to achieve highly generalizable policies.

## 2 BACKGROUND AND RELATED WORK

### 2.1 REINFORCEMENT LEARNING AND OFFLINE REINFORCEMENT LEARNING

The objective of RL is to learn a policy that maximizes the expected cumulative discounted reward in a Markov Decision Process (MDP) (Sutton & Barto, 2018). An MDP $M$ can be defined by a six-arity tuple $(\mathcal{S}, \mathcal{A}, T, r, \gamma, d_0)$ where $\mathcal{S}$ and $\mathcal{A}$ represent the state and action spaces respectively, $T(s'|s, a)$ and $r(s, a)$ represent the transition function and reward function of the dynamics, $\gamma \in [0, 1]$ denotes the discount factor, and $d_0$ denotes the distribution over initial states. In this paper, we let $R_{\max} = \max_{s,a} r(s, a)$ be the highest reward and assume $R_{\max} \geq 0$.

For a given policy $\pi$, its value function $V_M^\pi(s) = \mathbb{E}_{\pi,T}[\sum_{t=0}^\infty \gamma^t r(s_t, a_t)|s_0 = s]$ represents the expected cumulative discounted reward for the trajectories collected by policy $\pi$ starting from $s_0$ under MDP $M$. We define $d_M^\pi(s)$ to be the discounted occupancy measure of states for policy $\pi$ and MDP $M$, satisfying $d_M^\pi(s) = (1 - \gamma) \sum_{t=0}^\infty \gamma^t d_{t,M}^\pi$ where the occupancy of time step $t$ satisfying $d_{t,M}^\pi(s) = \sum_{s',a'} d_{t-1,M}^\pi(s')\pi(a'|s')T(s|s', a')$ and $d_{0,M}^\pi(s) = d_0(s)$. Based on the definition of state occupancy measure (Ho & Ermon, 2016), we further define the state-action distribution as $d_M^\pi(s, a) = d_M^\pi(s)\pi(a|s)$. The performance of policy $\pi$, which is the cumulative discounted reward under $M$, can be decomposed as the inner product of the state-action distribution and reward function: $\eta_M(\pi) = \mathbb{E}_{(s,a)\sim d_M^\pi}[r(s, a)]$. The goal of policy optimization, or RL, is to maximize its performance $\eta_M(\pi)$ in the given MDP $M$. The goal of offline RL, however, is to optimize the policy with a static

offline dataset $D$ and with no additional online interactions. This line of research often seeks the optimal policy within the support constraint of the offline dataset, rather than searching for the global optimal policy. The effectiveness of this approach depends highly on whether the offline dataset provides sufficient coverage of the state-action space in interest, which in turn limits its practical application since a substantially large amount of dataset can still be hard to collect.

## 2.2 Model-based Reinforcement Learning

To reduce the demand for interaction data, model-based RL algorithms extract dynamics models from interaction data to generate synthetic trajectories. However, the learned dynamics can be inaccurate at places where no interaction is available, and such error can accumulate over long-term planning (Asadi et al., 2019), thus negatively affecting subsequent policy optimization (Janner et al., 2019). In offline setting, various methods are proposed to mitigate this issue, such as reward penalty (Yu et al., 2020; Sun et al., 2023), modification to MDP (Kidambi et al., 2020; Yu et al., 2021), or adversarial training (Rigter et al., 2022), all of which align with the spirit of conservatism in light of uncertainty. Alternatively, MAPLE (Chen et al., 2021) proposes to learn a dynamics-aware policy on a bunch of model ensembles, which can identify and adapt to the real environment during test time. However, the model ensemble in MAPLE is generated via supervised training on the offline dataset, while in ReDM we are considering an extreme case where no or limited interaction data is available. Such a scenario issues a demand for the efficiency of the model learning process as we demonstrate in the next section.

## 2.3 Environment Generation

Environment generation is a commonly used technique that targets generating multiple MDPs which share similar components or structures to the target environments, to improve the generalization ability and robustness when deploying a policy. Several methods that have effectively improved zero-shot generalization can be classified as instances of environment generation. Among them, domain randomization (DR) (Peng et al., 2018; Tobin et al., 2017; Zhou et al., 2023; Kadokawa et al., 2023) perturbs the parameters of a simulator designed by domain experts. Another example is procedural content generation (PCG) (Cobbe et al., 2020; Küttler et al., 2020; Dutra et al., 2022; Agarwal & Shridevi, 2023), which assumes an overall generation structure of the environment and creates a range of MDPs by varying some pre-defined features according to random seeds.

In this paper, we are considering a more general scenario where neither such parameter ranges nor a high-fidelity and adjustable simulator is available. Recent research (Khirodkar et al., 2018; Rigter et al., 2022; Rigaki & Garcia, 2023) introduces adversarial training in environment generation, but this may result in over-pessimistic and unsolvable dynamics (Dennis et al., 2020; Kirk et al., 2023; Mediratta et al., 2023). While in ReDM, we also motivated to design our candidate tasks to effectively facilitate the optimization. However, both DR and PCG require the pre-defined parameters range of the environment or designed simulator to ensure that the generated environment is solvable.

## 3 Method

In this section, we introduce the ingredients of ReDM, which for the first time incorporates the idea of rehearsal into the RL process. In Section 3.1, we present the overall framework of policy rehearsing. In Section 3.2, we outline two principles of model generation, which we identify as keys to efficient candidate dynamics generation. In Section 3.3, we propose an iterative algorithm as a practical implementation. Lastly in Section 3.4, we explore ways to incorporate offline data to extend ReDM towards application in such scenarios.

## 3.1 Framework of Policy Rehearsing

In RL, the ultimate goal is to obtain a policy that performs well on a target MDP $M^* = \{\mathcal{S}, \mathcal{A}, T^*, r, \gamma, d_0)\}$. However, the target MDP is typically unavailable during the training process, and optimizing the policy in one inaccurate environment may lead to potential risk of mismatch. In policy rehearsing, we shift the focus to meta-optimizing the policy over a set of candidate dynamics models. Specifically, we assume the dynamics differ only in their transition function, and let the

---

**Algorithm 1** Framework of Policy Rehearsing

---

1: **Input:** Initial context-based policy $\pi_\theta$, context extractor $\phi_\psi$, and knowledge $\mathcal{K}$ about the environment (can be in any form).
2: Obtain the candidate model set $\mathcal{M}^c$ via knowledge $\mathcal{K}$.
3: Learn policy $\pi_\theta$ and context extractor $\phi_\psi$ over $\mathcal{M}^c$ via Algorithm 3 in Appendix B.

---

space $\mathcal{M} = \{(\mathcal{S}, \mathcal{A}, T_i, r, \gamma, d_0)\}_i$ contains all possible dynamics. The target MDP, $M^*$, is the actual MDP of interest and lies somewhere in the space $\mathcal{M}$. Although $M^*$ is not available directly for policy optimization, we can generate candidate models from $\mathcal{M}$ to serve as proxies for optimization.

Unlike previous works that rely on a generative structure or simulator, we consider generating dynamics models only based on simple underlying knowledge about the target environment, such as the reward function, terminal function, or a rough estimation of transition. Such knowledge helps to narrow down the whole hypothesis space $\mathcal{M}$ to a reasonable candidate set $\mathcal{M}^c$. For example, reward functions aid in distinguishing *good* trajectories from bad ones, terminal functions early-stop unrealistic trajectories, while the range of the state space prevents transitioning to unreachable states, thus effectively eliminating unrealistic trajectories. In our setting, we consider generating the transition models with such knowledge of reward and termination.

An adaptive policy $\pi^a$ is later trained on the set $\mathcal{M}^c$ to optimize the objective $\pi^a = \arg\max_\pi \sum_{M \in \mathcal{M}^c} \eta_M(\pi)$. In order to adapt to various dynamics models, we adopt a context-based policy to serve as the adaptive policy $\pi^a$, which consists of two components: a context extractor $\phi$ and context-dependent policy $\pi$, and a meta-training style pipeline to optimize the policy under different dynamics in the training stage. Then we obtain the formulation for adaptive policy optimization:

$$\mathcal{L}_{\mathrm{rl}} = \sum_{t=0}^{\infty} \gamma^t \mathbb{E}_{s_t \sim d_{t,M}^\pi, a_t \sim \pi(\cdot|s_t, z_t)}[r(s_t, a_t) + \mathcal{H}(\cdot|s_t)], \tag{1}$$

where $z_t = \phi(c_t)$ is context embedding of history $c_t = \{s_1, a_t, s_2, \cdots, a_{t-1}, s_t\}$. With a highly expressive adaptive policy formulation, we can achieve near-optimal control in all dynamics $M \in \mathcal{M}^c$ simultaneously. Thus, as long as the target $M^*$ could be roughly included in the set $\mathcal{M}^c$, the optimal control can be achieved in the target environment. We summarize the framework of policy rehearsing in Algorithm 1.

Random exhaustion of all models seems a way to construct $\mathcal{M}^c$. However, naively constructing $\mathcal{M}^c$ can be inefficient as the space for qualified dynamics searching is still too large. It is almost impossible to obtain all the models. Therefore, we turn to a more effective way by limited selection of models from the whole set, where the selected models, called candidate models, aim to significantly enhance the generalization of adaptive policy on the potential target environments. We discuss how to select representative candidate models in Section 3.2.

### 3.2 Principles of Model Generation

To efficiently generate candidate dynamics, we first analyze how to improve the performance of the learned adaptive policy. To begin with, we first characterize the gap between the candidate model set $\mathcal{M}^c$ and the target MDP $M^*$.

**Definition 3.1.** *(Optimal Policy Gap). Let all candidate dynamics in $\mathcal{M}^c$ be identical to the target MDP $M^*$ except for the transition function. The optimal policy gap $\epsilon_e$ is defined as the worst-case performance gap between the optimal performance in each $M \in \mathcal{M}^c$ and $M^*$:*

$$\max_{M \in \mathcal{M}^c} |\eta_{M^*}(\pi_{M^*}^*) - \eta_M(\pi_M^*)| \leq \epsilon_e,$$

*where $\pi_{M^*}^*$ is the optimal policy in $M^*$ and $\pi_M^*$ is the optimal policy in $M$.*

This discrepancy highlights the candidate model set's capability to derive a proficient policy in the learned model itself. It can be optimized if the ad-hoc optimal policy $\pi_M^*$ in each candidate $M$ is performant measured by the reward function. Instead of maintaining separate optimal policies for each candidate dynamics, ReDM employs a single adaptive policy $\pi^a$ to achieve near-optimal control in all dynamics. Consequently, it is necessary to consider the performance cost accompanying the adaptation. We here make a simplified and mild assumption about the cost during the adaptation process like (Chen et al., 2021),:

**Assumption 3.2.** *For any candidate MDP model $M$ from $\mathcal{M}^c$, the adaptive policy $\pi^a = \arg\max_\pi \sum_M \eta_M(\pi)$ satisfies*

$$\eta_M(\pi^a) \geq \eta_M(\pi_M^*) - \epsilon_a,$$

*where $\pi_M^*$ is the optimal policy in $M$ and $\epsilon_a$ is the adaptive cost which takes a positive value.*

Then we come to the following theorem, which describes the performance bound of policy rehearsing:

**Theorem 3.3.** *Given an MDP model set $\{M_i\}$ and its optimal adaptive policy $\pi^a = \arg\max_{\pi^a} \sum_{M_i \in \{M_i\}} \eta_{M_i}(\pi^a)$, if the target MDP $M^*$ satisfies $\min_i D_{\mathrm{TV}}(d_{M_i}^{\pi^a}, d_{M^*}^{\pi^a}) \leq \epsilon_m$, where $\epsilon_m > 0$, then we have:*

$$\eta_{M^*}(\pi^a) \geq \eta_{M^*}(\pi^*) - \epsilon_e - 2R_{\max}\epsilon_m - \epsilon_a,$$

*where $\pi^*$ is the optimal policy in $M^*$ and $D_{\mathrm{TV}}(d_{M_i}^\pi, d_{M^*}^\pi) = \frac{1}{2}\sum_{s,a}|d_{M_i}^\pi(s,a) - d_{M^*}^\pi(s,a)|$ denotes the total variance divergence.*

The proof can be found in Appendix A.3. This theorem highlights the factors which bound the performance loss when deploying the adaptive policy, namely $\epsilon_e$ and $\epsilon_m$. Here, $\epsilon_e$ is determined by the performance of the ad-hoc optimal policy on each dynamics. To control $\epsilon_e$, we need to ensure that the candidate model is capable of deriving a performant policy, i.e., to make sure that each candidate is eligible for policy optimization. Secondly, the term $\epsilon_m$ measures the minimum distance between the candidate model set $\mathcal{M}^c$ and the target MDP $M^*$. A practical strategy to control this term is to diversify the candidate set within our reach so that the target will fall into the vicinity of the candidates with higher probabilities. Finally, the term $\epsilon_a$ describes the cost of the meta-policy in identifying the dynamics characteristics. It is not solely dependent on the model generation process. To sum up the intuitions from Theorem 3.3, we propose two principles that should be taken into consideration when generating candidate dynamics: **diversity** and **eligibility**.

### 3.3 IMPLEMENTATIONS OF MODEL GENERATION

In this section we elaborate on how ReDM practically implements the aforementioned principles of model generation, including **diversity** and **eligibility**, where:

**(a) Diversity.** Quantifying the divergence between models can be challenging since environment dynamics is complex and high-dimensional in nature and we don't assume the generative structures or available probing data in ReDM. Inspired by recent advances in applying adversarial training (Goodfellow et al., 2016), ReDM leverages the performance gap of *current* policy to measure the divergence between models. Specifically, suppose at $k$-th iteration we have a candidate model set $\mathcal{M}_k^c$ and an adaptive policy $\pi_k^a$ which has been optimized over $\mathcal{M}_k^c$ such that $\pi_k^a = \arg\max_{\pi'} \sum_{M \in \mathcal{M}_k^c} \eta_M(\pi')$. If we manage to generate a new candidate dynamics $M'$ where $\pi_k^a$ performs poorly, then $M'$ should be distinct from previous candidate models $\mathcal{M}_k^c$ in both trajectory and single step level. The relationship between model divergence and policy performance can be characterized with Lemma 3.4.

**Lemma 3.4.** *Given a set of MDP models $\mathcal{M} = \{M_i\}_{i=1}^k$, policy $\pi = \arg\max_{\pi'} \sum_{i=1}^k \eta_{M_i}(\pi')$, and an MDP model $M_{k+1}$ satisfying $\min_{i\in\{1,\cdots,k\}} \eta_{M_i}(\pi) - \eta_{M_{k+1}}(\pi) \geq \delta$, then we have the occupancy discrepancy between $M_{k+1}$ and $\mathcal{M}$ that satisfies $\min_{i\in\{1,\cdots,k\}} D_{\mathrm{TV}}(d_{M_i}^\pi, d_{M_{k+1}}^\pi) \geq \frac{\delta}{2R_{\max}}$, and single-step discrepancy satisfies $\min_{i\in\{1,\cdots,k\}} \mathbb{E}_{s,a}[D_{\mathrm{TV}}(T_i(\cdot|s,a), T_{k+1}(\cdot|s,a))] \geq \frac{\delta(1-\gamma)}{2R_{\max}}$, where $T_i$ is the transition of the MDP model $M_i$.*

The derivation is provided in Appendix A.2. This motivates us to iterate between adaptive policy optimization and candidate dynamics generation, and for the $k$-th generation, our goal is to find a new dynamics model $M_{t+1}$ that satisfies: $M_{k+1} = \arg\min_M \eta_M(\pi_k^a)$. Note that when $\pi_k^a$ is fixed, this objective can be deemed as a standard RL objective where $M$ is the agent and $\pi_k^a$ serves as the environment. The objective can further be rewritten as

$$M_{k+1} = \arg\min_M \mathbb{E}_{(s,a)\sim d_M^{\pi_k^a}, s'\sim M}[r^c(s')], \tag{2}$$

where $r^c(s') = \mathbb{E}_{a'\sim\pi_k^a(\cdot|s')}[r(s',a')]$. The derivation is given in Appendix A.1. In principle, any RL algorithm can be employed to optimize this objective, with the dynamics model being treated as a distinct agent that needs to be learned.

**(b) Eligibility.** If a dynamics model is eligible, it should be able to produce viable paths toward high returns with a given reward function. To enhance eligibility, we encourage the generated candidate dynamics to transit to states from which a planning-based algorithm can obtain high returns, by adding state-dependent intrinsic reward defined as $r^e(s') = \max_{\tau_i(s')} R(\tau_i(s'))$, where $\{\tau_i(s')\}_{i=1}^N$ are random trajectories starting from $s'$ and $R(\tau_i(s')) = \sum_t \gamma^t r_t$ denotes the discounted cumulative reward of the trajectory. The reward is computed on-the-fly and utilized to optimize the generated dynamics models by an RL method. To sum up, our final objective for model generation is an RL objective involving both $r^c$ and $r^e$:

$$M_{k+1}^c = \arg\max_M \mathbb{E}_{(s,a)\sim d_M^{\pi_k^a}, s'\sim M}[-r^c(s') + \lambda r^e(s')], \tag{3}$$

where $\lambda$ balances diversity and eligibility. In our implementation, we employ Proximal Policy Optimization (PPO) (Schulman et al., 2017) to optimize this objective, and the pseudo-code of model generation in ReDM is presented in Algorithm 2 (see Appendix B).

### 3.4 MODEL GENERATION WITH OFFLINE DATA

As tasks become increasingly complex, the hypothesis space of dynamics models can catastrophically explode such that the generated candidate models cannot effectively approximate the target environment. In such circumstances, generating dynamics with the guidance of reward function solely can no longer suffice policy rehearsing. One relaxed assumption is the availability of pre-collected offline interaction data from the target or a misspecified but similar environment. Such data can help narrow down the hypothesis space of dynamics model since it provides knowledge about the ground truth transition of the target environment on state-action pairs covered by the offline dataset. In light of this, we extend the framework of ReDM by augmenting it with a regularization term to incorporate the offline dataset, leading to the following objective:

$$M_{k+1}^c = \arg\max_M \mathbb{E}_{(s,a)\sim d_M^{\pi_k^a}, s'\sim M}[-r^c(s') + \lambda r^e(s')] + \alpha\mathbb{E}_{(s,a,s')\sim D_\beta}[\log T(s'|s,a)], \tag{4}$$

where $D_\beta$ is the given offline dataset and $\alpha$ is the coefficient balancing RL and regularization. The regularization term resembles previous offline model-based RL algorithms, which extract an accurate model from the offline dataset. However in ReDM, we don't impose strict demands on the dataset, and the dataset can be small in amount or slightly mismatched in dynamics. Another remark is that we used a pre-trained policy as the planner to calculate the eligibility reward $r^e$. The extended algorithm is termed ReDM-o, and the pseudo-code is provided in Appendix B.

## 4 EXPERIMENTS

In this section, we aim to investigate the following questions: (1) Can policy rehearsing and ReDM facilitate policy optimization when no interaction data is available? (See Section 4.1). (2) As an environment generation method, can ReDM efficiently approximate the target environment? Are the building blocks of ReDM really necessary? (See Section 4.2). (3) Does ReDM-o, the ReDM extension with limited or mismatched data, aid policy optimization in complex tasks? (See Section 4.3).

### 4.1 REDM WITH NO INTERACTION DATA

To answer question (1), we conduct experiments on three representative Gym (Brockman et al., 2016) environments with continuous action spaces, namely InvertedPendulum, MountainCar (Continuous) and Acrobot. To validate ReDM, we instantiate each environment with five tasks by varying specific parameters of its simulator. For InvertedPendulum, the five tasks are created by setting the gravity coefficient to $0.5, 0.8, 1.0, 1.2$ and $1.5$ times the standard value of $-9.81$. For MountainCar, the five tasks differ in the angle of the mountain which is controlled by a parameter named $coef$. We set $coef = 2.0, 2.5, 3.0, 3.5$ and $4.0$, respectively, with the original value being $3.0$. For Acorbot, we change the frequency time of the 4th-order Runge-Kuta method to $0.15, 0.2, 0.25, 0.3$ and $0.4$, respectively. In the model generation process, we only utilize the true reward function, the terminal function, and we sample the initial states for rollouts from a Gaussian distribution $\mathcal{N}(\mathbf{0}, \mathbf{I})$. We test ReDM with 5 independent runs for each task and record the means and standard deviations. Figure 1 illustrates the relative performance of ReDM against a random policy, and it can be observed that

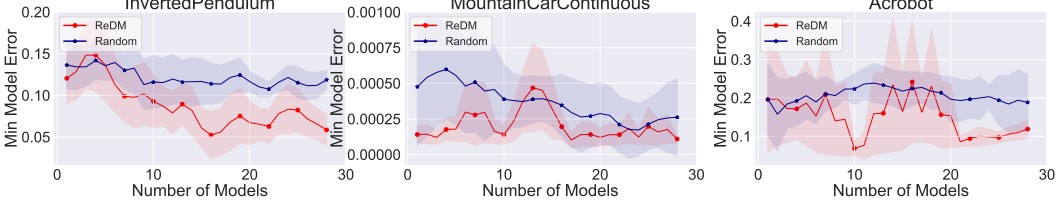

Figure 1: Relative performance between ReDM and the random policy on different environments with different hyper-parameters.

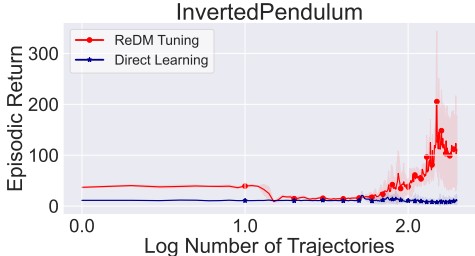

Figure 2: Model loss of ReDM on different environments. All results are averaged across 5 seeds.

ReDM significantly outperforms the random policy when no interaction data is available, indicating the effectiveness of policy rehearsing and ReDM. We also provide the learning curves in Appendix F. To illustrate the adaptation ability of the final learned policy, we fine-tune it with a few online trajectories and compare its performance to learning from scratch with the same policy architecture and the same amount of data The learning curves are plotted in Figure 3. Within a few trajectories, our learned adaptive policy could make a quicker adaptation than directly learning.

## 4.2 A CLOSER LOOK AT THE MODEL GENERATION IN REDM

In this section we will check each building block of ReDM to verify where improvements stem from. We start with introducing *minimal model error* $\Delta_{s,a}^{\mathcal{M}}$, which is defined as the minimal error achieved by a set of models $\mathcal{M}$, i.e., $\Delta_{s,a}^{\mathcal{M}} := \min_{M \in \mathcal{M}^c} \|M(s,a) - s'\|^2$ where $s'$ is the ground-truth transition of the target environment and $M(s,a)$ is the predicted next state by MDP model $M$. To track minimal model error, at the end of each iteration, we collect data by the learned policy in the target environment and report the average minimal model error $\bar{\Delta}_{s,a}^{\mathcal{M}}$ over the collected data. As shown in Figure 2, with more candidate models generated

Figure 3: Learning curves of ReDM tuning and direct online learning on InvertedPendulum. All results are averaged across 5 seeds.

and included into the set $\mathcal{M}^c$, the average error $\bar{\Delta}_{s,a}^{\mathcal{M}^c}$ gradually decreases, indicating that the set of candidate dynamics more closely approximates the target environment. This decrease in minimal model error in turn results in improved performance. Take MountainCar (Continuous) as an example, the minimal model error (orange line) reaches its lowest with approximately 20 models, which corresponds to roughly 200 gradient steps at which the performance of the adaptive policy begins to improve. To provide a comparison, we generate another candidate model set $\mathcal{M}^{\text{random}}$ by iteratively adding random dynamics models. However, we found that the minimum model error of this set (red line in the figure) is larger than that of ReDM with a non-negligible gap.

Then we visualize the information of trajectories generated by learned models. For the InvertedPendulum task, since we have generated dynamics models and learned the policy correspondingly, we use the learned policy to rollout in the real environment (1.0 times gravity) and the generated candidate models respectively. We select three models, including the most accurate one and two randomly selected models. We plot the angles and positions in the states changed over time steps in Figure 4(a) and 4(b). The results indicate that although the other generated models are quite distinguished from the real environment, there exists one model that is similar to the target environment.

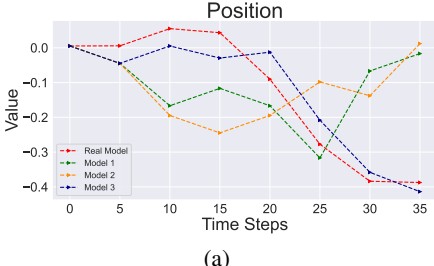
(a)

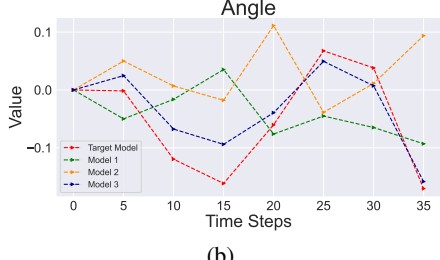
(b)

Figure 4: The data rollout by timesteps from distinguished generated models and target environments in InvertedPendulum. (a) is the position information of the data. (b) is the angle information of the data. There exists a model (red line) that could generate similar data as real environment (blue line).

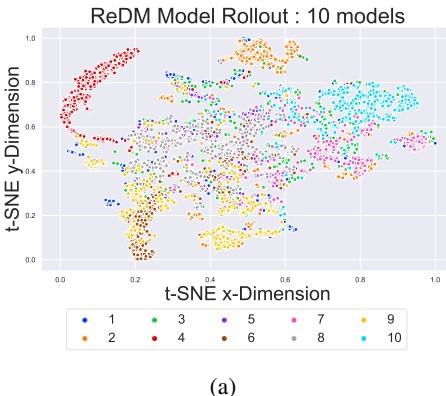
(a)

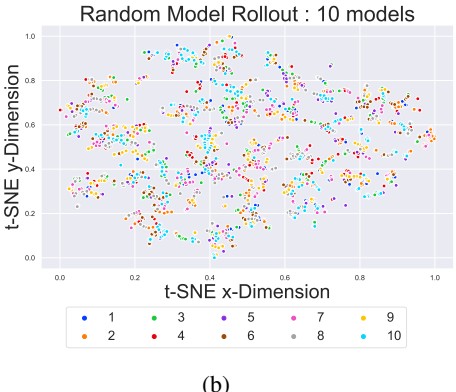
(b)

Figure 5: The t-SNE results of the data rollout from distinguished generated models. We use the final checkpoint of the adaptive policy to roll out on ten candidate models in the candidate set and collect the interaction data. The data are then reduced into two-dimension via t-SNE. (a) is the projected data from our generated models, and (b) is the projecte

To illustrate the diversity of the candidate model set generated by ReDM, we employ t-SNE (Van der Maaten & Hinton, 2008) to visualize the distribution of the interaction data produced by each candidate model in Figure 5(a). Similarly we present the t-SNE of $\mathcal{M}^{random}$ in Figure 5(b). It is evident from the plots that the data manifolds of each candidate generated by ReDM are clearly grouped into different clusters, which means ReDM generates more distinct dynamics and provides more benefits for policy rehearsing compared to the baseline.

Lastly, we ablate the design of diversity and eligibility. As a preliminary check, we run ReDM on InvertedPendulum without the diversity reward or the eligibility reward. We found that neither of them yields a valid policy. To identify the root cause of the degeneration, we calculate the accumulated return of

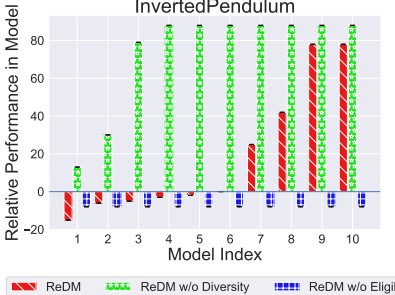

Figure 6: The sorted relative performance between the first 10 generated models and target environment in each policy, where the red column is the model of ReDM, the green is that without diversity, and the blue is ReDM without eligibility.

the meta-policy given by (1) ReDM, (2) ReDM without diversity, and (3) ReDM without eligibility in their generated models. Specifically, upon the generation of the 10th candidate model, we use the policy at that iteration to interact with each candidate model to obtain the performances of the $i$-th candidate, $i \in [10]$. Later we rollout the same policy in the ground truth environment to obtain the reference performance. The relative performance is calculated as the $i$-th performance minus the reference performance and reported in Figure 6. We sort the indices of candidate models according to their relative performance. The result reveals that (1) without eligibility, candidate models tend to be overly pessimistic, resulting in a significantly lower accumulated return than the ground truth; (2) without diversity, the candidate models can be overly optimistic, as the policy achieves much higher performance in all candidate models; (3) by incorporating both diversity and eligibility constraints, the evaluations of the policy are diversified and there even exists one model (e.g., model 3 in Figure 6) whose evaluation is similar to that in the target environment.

### 4.3 ReDM-o with Interaction Data

In this section, we aim to test ReDM-o, which incorporates external offline data or demonstrations in more complex tasks to answer the question (3). Our evaluations in this section differ from standard offline RL in that we test on offline data limited in both quantity and quality, or data collected from mismatched environments. Standard offline

Table 1: Performance of ReDM-o and baselines on D4RL with limited data. All scores presented are averaged over 5 seeds and normalized by the way proposed by (Fu et al., 2020). We bold the highest mean. Hyper-parameters can be found in Appendix 3.

| Type | Data Size | ReDM-o | MB-best | MF-best |
|------|-----------|--------|---------|---------|
| HalfCheetah-Random | 200 | **2.5 ± 0.9** | 2.2 | 0.6 |
| HalfCheetah-Random | 5000 | **19.0 ± 1.5** | 9.3 | 9.5 |
| Hopper-Random | 200 | **23.8 ± 8.0** | 14.3 | 1.1 |
| Hopper-Random | 5000 | **31.4 ± 0.3** | 13.8 | 2.5 |

RL methods may be compromised in these settings, as the coverage, quality and unbiasedness of the data are vital for policy optimization. However, we expect ReDM-o to have a better performance as ReDM-o does not rely solely on the provided data. We compare ReDM-o with several offline RL methods, such as model-free methods including **CQL** (Kumar et al., 2020), **TD3BC** (Fujimoto & Gu, 2021), **IQL** (Kostrikov et al., 2022), and model-based methods including **MOPO** (Yu et al., 2020) and **MAPLE** (Chen et al., 2021). Here MAPLE is a model-based method that generates an ensemble of dynamics models via supervised learning and also employs an adaptive policy to meta-train on the generated dynamics. As it only involves supervised learning, it can be a direct ablation of the model generation designs in ReDM-o. We report the best results among the model-free baselines (**MF-best**) and model-based baselines (**MB-best**).

For the first experiment, we sample a subset of data, which is only 200 or 5000 transitions, from random datasets in D4RL and feed the samples to each algorithm. Results are presented in Table 1. The performance of standard offline RL algorithms significantly deteriorates when data is scarce and of poor quality, while ReDM-o achieves higher performance compared to the baselines, including MAPLE, which validates the effectiveness of our model generation process. To comprehensively evaluate the generalization ability, we test ReDM-o with full datasets from D4RL, including domains like *HalfCheetah, Hopper, Walker2D* and dataset qualities like *random, medium, medium-replay and medium-expert*. We further evaluate the optimized meta-policies in mismatched dynamics, by multiplying the coefficient of gravity by 0.5, 1.0, and 1.5. We report the scores averaged over all tasks and all gravity factors in Figure 7. ReDM-o is still performant compared to other baselines. For more detailed information, such as the performance of the ReDM-o and baselines on each task and gravity coefficient, please refer to Appendix E.

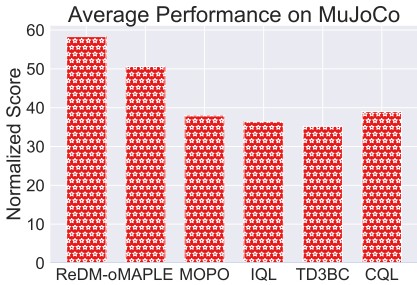

Figure 7: Averaged performance of ReDM-o and baselines, which are evaluated on all the dataset of D4RL. For each dataset, methods are evaluated on environment, where the gravity changed by multiplying 0.5, 1.0, 1.5. All scores presented are averaged over 5 seeds and normalized as (Fu et al., 2020). Hyper-parameters can be found in Appendix 3.

## 5 Conclusion and Future Work

In this paper, we aim to optimize policies in scenarios where interactions or rich offline data are not available. Our method, Policy Rehearsing via Dynamics Model Generation (ReDM), incorporates the idea of rehearsal into reinforcement learning and generates candidate dynamics models for policy rehearsing. To scale to practical tasks, we establish principles of diversity and eligibility for model generation and also discuss how to leverage offline demonstrations to further narrow down the hypothesis space of models. We tested ReDM with three different settings: learning with no data, with a limited amount of data, and with mismatched data, and ReDM demonstrates consistent improvement in these scenarios. This work offers a fresh perspective on reevaluating reinforcement learning and dynamics model generation. While our method primarily concentrates on generating candidate models, the exploration of adaptive policy is still an avenue that requires further investigation.

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

## A PROOF

**Lemma A.1.** *For the objective $M = \arg\min_{M'} \eta_{M'}(\pi)$, where $\eta_M(\pi) = \mathbb{E}_{(s,a)}[r(s,a)]$, and $T$ is the transition of $M$, the objective is equal to*

$$M = \arg\min_M \mathbb{E}_{(s,a)\sim d_M^\pi, s'\sim T}[r^c(s')],$$

*where $r^c(s') = \mathbb{E}_{a'\sim\pi(\cdot|s')}[r(s',a')]$.*

*Proof.* For the objective $\eta_M(\pi)$, we have

$$\eta_M(\pi) = \mathbb{E}_{(s,a)\sim d_M^\pi}[r(s,a)]$$

$$= (1-\gamma)\sum_{t=0}^{\infty}\gamma^t \mathbb{E}_{(s,a)\sim d_{t,M}^\pi}[r(s,a)]$$

$$= (1-\gamma)\mathbb{E}_{(s,a)\sim d_0}[r(s,a)] + (1-\gamma)\sum_{t=1}^{\infty}\gamma^t \mathbb{E}_{(s',a')\sim d_{t-1,M}^\pi, s\sim T(\cdot|s,a), a\sim\pi(\cdot|s)}[r(s,a)]$$

$$= (1-\gamma)\mathbb{E}_{(s,a)\sim d_0}[r(s,a)] + (1-\gamma)\sum_{t=1}^{\infty}\gamma^t \mathbb{E}_{(s',a')\sim d_{t-1,M}^\pi, s\sim T(\cdot|s,a)}[\mathbb{E}_{a\sim\pi(\cdot|s)}[r(s,a)]]$$

$$= (1-\gamma)\mathbb{E}_{(s,a)\sim d_0}[r(s,a)] + (1-\gamma)\sum_{t=1}^{\infty}\gamma^t \mathbb{E}_{(s',a')\sim d_{t-1,M}^\pi, s\sim T(\cdot|s,a)}[r^c(s')]$$

$$= (1-\gamma)\mathbb{E}_{(s,a)\sim d_0}[r(s,a)] + \gamma(1-\gamma)\sum_{t=0}^{\infty}\gamma^t \mathbb{E}_{(s',a')\sim d_{t,M}^\pi, s\sim T(\cdot|s,a)}[r^c(s')]$$

$$= \underbrace{(1-\gamma)\mathbb{E}_{(s,a)\sim d_0}[r(s,a)]}_{a} + \underbrace{\gamma\mathbb{E}_{(s,a)\sim d_M^\pi, s'\sim T}[r^c(s')]}_{b},$$

where only the part (b) is related with the transition model as the part (a) is only with the initial distribution. Thus, minimizing $\eta_M(\pi)$ is equal to minimizing part (b). $\qquad\square$

**Lemma A.2** (Lemma 3.4). *Given a set of MDP models $\mathcal{M} = \{M_i\}_{i=1}^{k}$, policy $\pi = \arg\max_{\pi'}\sum_{i=1}^{k}\eta_{M_i}(\pi')$, an MDP model $M_{k+1}$ satisfying $\min_{i\in\{1,\cdots,k\}}\eta_{M_i}(\pi) - \eta_{M_{k+1}}(\pi) \geq \delta$, where $\delta$ is the performance gap, we have the cumulative discrepancy between the new model and the model set satisfying*

$$\min_{i\in\{1,\cdots,k\}} D_{\mathrm{TV}}(d_{M_i}^\pi, d_{M_{k+1}}^\pi) \geq \frac{\delta}{2R_{\max}}$$

*and single step discrepancy satisfying*

$$\min_{i\in\{1,\cdots,k\}} \mathbb{E}_{s,a}[D_{\mathrm{TV}}(T_i(\cdot|s,a), T_{k+1}(\cdot|s,a))] \geq \frac{\delta(1-\gamma)}{2R_{\max}},$$

*where $T_i$ is the transition of MDP model $M_i$ and $D_{\mathrm{TV}}(d_{M_1}^\pi, d_{M_2}^\pi) = \frac{1}{2}\sum_{s,a}|d_{M_1}^\pi(s,a) - d_{M_2}^\pi(s,a)|$ is the total variance divergence.*

*Proof.* First, we have

$$\min_i \eta_{M_i}(\pi) - \eta_{M_{k+1}}(\pi) = \min_i \sum_{s,a} r(s,a)|d_{M_i}^\pi - d_{M_{k+1}}^\pi|,$$

$$\leq \min_i \sum_{s,a} R_{\max}|d_{M_i}^\pi - d_{M_{k+1}}^\pi|,$$

then as $\min_i \eta_{M_i}(\pi) - \eta_{M_{k+1}(\pi)} \geq \delta$, we have

$$\min_i \sum_{s,a} R_{\max}|d_{M_i}^\pi - d_{M_{k+1}}^\pi| \geq \delta$$

$$\min_i \frac{1}{2} \sum_{s,a} R_{\max}|d_{M_i}^\pi - d_{M_{k+1}}^\pi| \geq \frac{\delta}{2}$$

$$\min_i \frac{1}{2} \sum_{s,a} |d_{M_i}^\pi - d_{M_{k+1}}^\pi| \geq \frac{\delta}{2R_{\max}}$$

$$\min_i D_{\mathrm{TV}}(d_{M_i}^\pi, d_{M_{k+1}}^\pi) \geq \frac{\delta}{2R_{\max}}.$$

For the single step transition, from the theoretical results in (Xu et al., 2020b), the relationship between the TV divergence of two transitions of model $M_1$ and $M_2$ and their occupancy measure under model $\hat{M}$ is

$$D_{\mathrm{TV}}(d_{M_1}^\pi, d_{M_2}^\pi) \leq \frac{\mathbb{E}_{s,a}[D_{\mathrm{TV}}[T_1(\cdot|s,a), T_2(\cdot|s,a)]]}{(1-\gamma)},$$

where $T_1$ and $T_2$ are the transitions of $M_1$ and $M_2$. Finally we have

$$\min_i \mathbb{E}_{s,a}[D_{\mathrm{TV}}(T_i(\cdot|s,a), T_{k+1}(\cdot|s,a))] \geq \frac{\delta(1-\gamma)}{2R_{\max}}.$$

The proof is completed. $\qquad\square$

**Lemma A.3** (Theorem 3.3). *Given a set of MDP model $\{M_i\}$ and its optimal adaptive policy $\pi^a = \arg\max_{\pi^a} \sum_{M_i \in \{M_i\}} \eta_{M_i}(\pi^a)$, if the target MDP model $M^*$ satisfies $\min_i D_{\mathrm{TV}}(d_{M_i}^{\pi^a}, d_{M^*}^{\pi^a}) \leq \epsilon_m$, where $\epsilon_m$ is a positive value, we have*

$$\eta_{M^*}(\pi^a) \geq \eta_{M^*}(\pi^*) - \epsilon_e - 2R_{\max}\epsilon_m - \epsilon_a,$$

*where $\pi^*$ is the optimal policy in $M^*$.*

*Proof.* For the distance condition $\min_i D_{\mathrm{TV}}(d_{M_i}^{\pi^a}, d_{M^*}^{\pi^a}) \leq \epsilon_m$, we have

$$\min_i \eta_{M_i}(\pi^a) - \eta_{M^*}(\pi^a) = \min_i \sum_{s,a} r(s,a)|d_{M_i}^\pi - d_{M^*}^\pi|$$

$$\leq \min_i \sum_{s,a} R_{\max}|d_{M_i}^\pi - d_{M^*}^\pi|$$

$$= \min_i 2R_{\max}D_{\mathrm{TV}}(d_{M_i}^\pi, d_{M^*}^\pi)$$

$$= 2R_{\max}\epsilon_m.$$

Then by the assumption, we have

$$\eta_{M_i}(\pi^a) \geq \eta_{M_i}^* - \epsilon$$
$$\geq \eta_{M^*}(\pi^*) - \epsilon_e - \epsilon_a,$$

where $\pi^*$ is the optimal policy in the model $M$.

Then $\forall i$ that $\eta_{M_i}(\pi^a) \geq \eta_{M^*}(\pi^*) - \epsilon_e - \epsilon_a$ and $\exists i$ that $\eta_{M_i}(\pi^a) \leq \eta_{M^*}(\pi^a) + 2R_{\max}\epsilon_m$, finally we have

$$\eta_{M^*}(\pi^a) \geq \eta_{M^*}(\pi^*) - \epsilon_e - 2R_{\max}\epsilon_m - \epsilon_a.$$

The proof is completed. $\qquad\square$

Then we provide a simple analysis for convergence:

**Lemma A.4.** *Given a set of MDP models $\mathcal{M}_k^c = \{M_i\}_{i=1}^k \subset \mathcal{M}$, policy $\pi_k = \arg\max_{\pi'} \sum_{i=1}^k \eta_{M_i}(\pi')$, and an MDP model $M_{k+1} \in \mathcal{M}$ satisfying $\min_{i\in\{1,\cdots,k\}} \eta_{M_i}(\pi_k) - \eta_{M_{k+1}}(\pi_k) = \delta_k$ and $M_{k+1} = \arg\min_{M\in\mathcal{M}} \eta_M(\pi_k)$, if we assume that $\eta_M(\pi_{k+1}) - \eta_M(\pi_k) = 0, \forall k, \forall M \in \mathcal{M} \setminus \mathcal{M}_{k+1}^c$ and $\min_{i\in\{1,\cdots,k\}} \eta_{M_i}(\pi_k) - \min_{i\in\{1,\cdots,k+1\}} \eta_{M_i}(\pi_{k+1}) \geq 0$, which implies an increasing adaptive cost by $k$, then for any $\epsilon > 0$, there must exists a certain $K$, for any $k > K$ we have $\delta_k < \epsilon$.*

---

**Algorithm 2** Candidate Model Generation

---

1: **Input:** Context-based policy $\pi_\theta$ parameterized by $\theta$ and context extractor $\phi_\psi$ parameterized by $\psi$.
2: Initialize a new candidate model $M$, with its transition function $T_\mu$ parameterized by $\mu$.
3: Initialize an empty data buffer $D_{\mathrm{model}} = \emptyset$.
4: **for** $i = 1$ **to** $E_{\mathrm{model}}$ **do**
5:     Sample initial state $s_0$.
6:     **for** $t = 0$ **to** $H$ **do**
7:         Sample action $a_t \sim \pi_\theta(\cdot|s_t, \phi_\psi(c_t))$, where $c_t$ is the context containing history transitions.
8:         Sample next state $s_{t+1} \sim T_\mu(\cdot|s_t, a_t)$ and calculate the reward $r_t^c$.
9:         Sample $N$ trajectories from $s_{t+1}$ with random policy to compute eligible reward $r_t^e(s_{t+1})$ by Eq. 2.
10:        Mix reward $-r_t^c$ with $r_t^e$ by Eq. 3.
11:        Add data to model buffer $D_{\mathrm{model}}$.
12:     **end for**
13:     Update $\mu$ with model buffer by PPO.
14:     Reset the data buffer $D_{\mathrm{model}} = \emptyset$.
15: **end for**
16: **Return** candidate model $M$.

---

*Proof.* First, we have

$$
\begin{aligned}
\min_{i \in \{1,\cdots,k\}} \eta_{M_i}(\pi_k) - \eta_{M_{k+1}}(\pi_k) &\geq \min_{i \in \{1,\cdots,k\}} \eta_{M_i}(\pi_k) - \eta_{M_{k+2}}(\pi_k) \\
&\geq \min_{i \in \{1,\cdots,k\}} \eta_{M_i}(\pi_k) - \eta_{M_{k+2}}(\pi_{k+1}) \\
&\geq \min_{i \in \{1,\cdots,k\}} \eta_{M_i}(\pi_k) - \min_{i \in \{1,\cdots,k+1\}} \eta_{M_i}(\pi_{k+1}) \\
&\quad + \min_{i \in \{1,\cdots,k+1\}} \eta_{M_i}(\pi_{k+1}) - \eta_{M_{k+2}}(\pi_{k+1}) \\
&\geq \min_{i \in \{1,\cdots,k+1\}} \eta_{M_i}(\pi_{k+1}) - \eta_{M_{k+2}}(\pi_{k+1})
\end{aligned}
$$

Thus, we have $\delta_k \geq \delta_{k+1}$. And when we have all the models in our candidate set, $\delta_k$ should be zero. $\delta_k$ will converge to zero by the increasing of $k$.

$\square$

The theorem demonstrates that as the number of generated models increases, it becomes increasingly difficult to find models that perform poorly within the hypothesis space. This implies that our policy gradually adapts to the entire hypothesis space.

## B   ALGORITHMS

Algorithms 2∼5 are the pseudo-codes of candidate model generation, policy optimization, candidate model generation with offline data, ReDM and ReDM-o, respectively.

Algorithm 2 describes the process of generating one candidate model. First, we initialize a parameterized dynamics model $T_\mu$ as the candidate model $M$. In lines 2-10, we rollout the current meta-policy $\pi_\theta$ on the candidate model $M$. At each timestep, the meta-policy selects actions based on the state $s_t$ and the context feature $z_t = \phi_\psi(s_t)$, and the model predicts the next state $s_{t+1}$ and the immediate reward $r_t^c$. In line 7, we use a random shooting planner to compute the eligibility reward $r_t^e$. After the data collection, we employ Proximal Policy Optimization (PPO) to optimize the candidate model (line 11). This process will repeat for $E_{\mathrm{model}}$ epochs, after which the new candidate will be returned.

Algorithm 3 summarizes the optimization of the meta-policy. The algorithm takes up-to-date meta policy $\pi_\theta$, the associated context encoder $\phi_\psi$, and the current candidate model set $\mathcal{M}^c$ as input. During each update epoch, we randomly select a candidate model from the set and rollout the

---

**Algorithm 3** Policy Optimization

---

1: **Input:** Context-based policy $\pi_\theta$ and context extractor $\phi_\psi$, candidate model set $\mathcal{M}^c$.
2: Initialize the data buffer $D_{\mathrm{policy}} = \emptyset$.
3: **for** $i = 1$ **to** $E_{\mathrm{policy}}$ **do**
4:     Randomly sample a model $M^c$ from $\mathcal{M}^c$ as current model. Denote $T^c$ as its dynamics function, and sample initial state $s_0$.
5:     **for** $t = 0$ **to** $H$ **do**
6:         Sample action $a_t \sim \pi_\theta(\cdot|s_t, \phi_\psi(c_t))$, where $c_t$ is the context containing history transitions.
7:         Sample next state $s_{t+1} \sim T^c(\cdot|s_t, a_t)$ and calculate the reward $r_t$.
8:         Add reward penalty to $r_t$ based on the uncertainty of models.
9:         Add $(s_t, a_t, s_{t+1}, r_t)$ to dataset $D_{\mathrm{policy}}$.
10:     **end for**
11:     Update $\theta$ and $\psi$ with $D_{\mathrm{policy}}$ using SAC.
12: **end for**
13: **Return** context-based policy $\pi_\theta$ and context extractor $\phi_\psi$

---

**Algorithm 4** ReDM

---

1: Initialize the context-based policy $\pi_\theta$ and the context encoder $\phi_\psi$.
2: Initialize the candidate model set $\mathcal{M}^c = \emptyset$.
3: **for** $k = 1$ **to** $E_{\mathrm{total}}$ **do**
4:     Generate a new candidate dynamics model $M$ via Algorithm 2.
5:     Let $\mathcal{M}^c \leftarrow \mathcal{M}^c \cup \{M\}$.
6:     Update policy $\pi_\theta$ and context extractor $\phi_\psi$ via Algorithm 3.
7: **end for**

---

meta-policy to collect data (line 3-8), with which we update the meta-policy and the encoder using SAC (line 9). The optimization continues for $E_{\mathrm{policy}}$ epochs.

The pseudo-code for ReDM is outlined in Algorithm 4. ReDM iteratively performs policy optimization and candidate model generation. In each iteration, it will call Algorithm 2 to generate a new candidate dynamics model, followed by meta-policy optimization by Algorithm 3.

For ReDM-o, it differs slightly from ReDM only in the model generation process due to incorporating the offline data. We list the pseudo-code for candidate model generation in Algorithm 5. Major differences are (1) the model is pre-trained to fit the offline data after initialization, and further updated via Eq. 4 in the loop; (2) we used a pre-trained policy $\pi_b$ in replace of the random planner for efficiency. The policy $\pi_b$ is trained via SAC with the offline data, plus an additional BC regularize similar to TD3+BC (Fujimoto & Gu, 2021); (3) the initial states are sampled from offline data. Apart from these, the offline dataset is also involved in policy optimization, as we follow existing practices (Yu et al., 2020) to train the meta-policy with a mixture of offline data and synthetic rollouts.

## C   Details of Experiments

In this section, we introduce the baselines and provide details for how we obtained the results in Section 4.

### C.1   Baseline Methods

**IQL** (Kostrikov et al., 2022). Implicit Q-Learning aims to mitigate errors caused by out-of-distribution data by focusing on estimating the Q-value and improving the policy solely within the in-distribution data. The algorithm achieves this through two main steps: value iteration using expectile regression and policy learning using advantage-weighted regression. During value iteration, IQL employs expectile regression to approximate the in-sample optimal state value, by assigning higher weights to samples with positive advantages. The policy optimization behavior clones offline data with a weight of exponent of the advantage. The whole process eliminates the risk of incorporating out-of-dataset actions, thus providing a more safe and robust improvement.

---

**Algorithm 5** Candidate Model Generation with Offline Data

---

1: **Input:** Context-based policy $\pi_\theta$ parameterized by $\theta$ and context extractor $\phi_\psi$ parameterized by $\psi$. Offline data buffer $D_{\text{offline}}$ and a pre-trained policy $\pi_b$ for planning.
2: Initialize a new candidate model $M$, with its transition function $T_\mu$ parameterized by $\mu$. Fit $T_\mu$ with the offline data for several epochs.
3: Initialize an empty data buffer $D_{\text{model}} = \emptyset$.
4: **for** $i = 1$ **to** $E_{\text{model}}$ **do**
5:    Sample initial state $s_0$ from $D_{\text{offline}}$.
6:    **for** $t = 0$ **to** $H$ **do**
7:       Sample action $a_t \sim \pi_\theta(\cdot|s_t, \phi_\psi(c_t))$, where $c_t$ is the context containing history transitions.
8:       Sample next state $s_{t+1} \sim T_\mu(\cdot|s_t, a_t)$ and calculate the reward $r_t^c$.
9:       Sample $N$ trajectories from $s_{t+1}$ with $\pi_b$ to compute eligible reward $r_t^e(s_{t+1})$ by Eq. 2.
10:      Mix reward $-r_t^c$ with $r_t^e$ by Eq. 3.
11:      Add data to model buffer $D_{\text{model}}$.
12:    **end for**
13:    Update $\mu$ with $D_{\text{model}}$ and $D_{\text{offline}}$ by Equation 4.
14:    Reset the data buffer $D_{\text{model}} = \emptyset$.
15: **end for**
16: **Return** candidate model $M$.

---

**TD3BC** (Fujimoto & Gu, 2021). Twin Delayed Deep Deterministic Policy Gradient with Behavior Cloning combines the TD3 (Twin Delayed Deep Deterministic Policy Gradient) and behavior cloning techniques. TD3BC incorporates delayed updates for the target Q-networks, which further stabilizes the learning process. For policy improvement, TD3BC additionally incorporates a regularization term which forces the policy to imitate actions in offline data.

**CQL** (Kumar et al., 2020). Conservative Q-Learning addresses the problem of overestimation in Q-learning methods and provides a conservative estimate of the Q-values to ensure cautious policy updates. It achieves this by adding a regularization term to the standard Q-learning objective. This term penalizes OOD actions with high Q-values. By doing so, CQL encourages the agent to be more conservative and avoids overly optimistic estimates.

**MOPO** (Yu et al., 2020). Model-Based Offline Policy Optimization is an algorithm that utilize the dynamics model to learn a more generalizable policy than model-free methods. The algorithm consists of two main steps: model learning and policy optimization. In the model learning phase, MOPO trains a dynamics model using the offline dataset, which is used to generate synthetic rollouts. Later, MOPO combines the offline data and the synthetic rollouts for policy optimization. To account for model bias or model error, MOPO penalizes the reward of the synthetic rollouts based on the uncertainty of the models.

**MAPLE** (Chen et al., 2021). Model-Based Adaptive Policy Learning is a reinforcement learning algorithm that combines dynamics models and context-based policies to learn adaptive policies in complex tasks. In MAPLE, the model learning process is the same as that in other model-based methods like MOPO, but it introduces an RNN policy to fit different dynamics of ensemble models. With the RNN policy, MAPLE can adaptively handle the OOD area in the target environment.

**Code Sources**. For model-free offline algorithms including TD3BC, CQL and IQL, we use implementations from CORL (Tarasov et al., 2022), and keep the hyperparameter identical to the original papers. For MOPO and MAPLE, we used the implementation from the OfflineRL codebase [1]. All parameters are kept the same as their original papers.

## C.2 REDM

The hyperparameters we used are listed in Table 2 (for ReDM) and Table 3 (for ReDM-o). The hyper-parameters of concern include:

- $\lambda$, which balances between the diversity reward and eligibility reward in Eq. 3.

---

[1] `https://github.com/polixir/OfflineRL`

Table 2: Hyper-parameters used in experiments without interaction data.

| | Model Generation | | Policy Optimization | | Others | | |
| --- | --- | --- | --- | --- | --- | --- | --- |
| **Environment** | $\lambda$ | $N$ | **Penalty** | $H$ | $E_{\text{model}}$ | $E_{\text{policy}}$ | $E_{\text{total}}$ |
| InvertedPendulum | 0.2 | 50 | - | 20 | 200 | 20 | 25 |
| MountainCarContinous | 0.2 | 50 | - | 20 | 200 | 20 | 25 |
| AcRobot | 0.2 | 50 | - | 20 | 200 | 20 | 25 |

Table 3: Hyperparameters used in ReDM-o experiments with D4RL datasets.

| | | Model Generation | | | Policy Optimization | | Others | | |
| --- | --- | --- | --- | --- | --- | --- | --- | --- | --- |
| **Environment** | **Type** | $\lambda$ | $\alpha^*$ | $N$ | **Penalty** | $H$ | $E_{\text{model}}$ | $E_{policy}$ | $E_{total}$ |
| HalfCheetah | random | 0.1 | 0.01 | 50 | 0.25 | 20 | 5 | 50 | 40 |
| HalfCheetah | medium | 0.1 | 0.01 | 50 | 0.25 | 20 | 5 | 50 | 40 |
| HalfCheetah | medium-replay | 0.1 | 0.01 | 50 | 0.25 | 20 | 5 | 50 | 40 |
| HalfCheetah | medium-expert | 0.1 | 0.01 | 50 | 0.5 | 20 | 5 | 50 | 40 |
| Hopper | random | 0.1 | 0.01 | 50 | 0.25 | 20 | 5 | 50 | 20 |
| Hopper | medium | 0.1 | 0.01 | 50 | 0.25 | 20 | 5 | 50 | 20 |
| Hopper | medium-replay | 0.1 | 0.01 | 50 | 0.25 | 20 | 5 | 50 | 20 |
| Hopper | medium-expert | 0.1 | 0.01 | 50 | 0.25 | 20 | 5 | 50 | 20 |
| Walker2d | random | 0.1 | 0.01 | 50 | 0.25 | 10 | 5 | 50 | 20 |
| Walker2d | medium | 0.1 | 0.01 | 50 | 0.25 | 20 | 5 | 50 | 20 |
| Walker2d | medium-replay | 0.1 | 0.01 | 50 | 0.25 | 10 | 5 | 50 | 20 |
| Walker2d | medium-expert | 0.1 | 0.01 | 50 | 0.25 | 20 | 5 | 50 | 20 |
| Pen | cloned | 0.1 | 0.01 | 50 | 0.1 | 10 | 5 | 50 | 10 |
| Pen | human | 0.1 | 0.01 | 50 | 0.1 | 10 | 5 | 50 | 10 |
| Hammer | cloned | 0.1 | 0.01 | 50 | 0.1 | 10 | 5 | 50 | 10 |
| Hammer | human | 0.1 | 0.01 | 50 | 0.1 | 10 | 5 | 50 | 10 |

- $N$, which is the number of trajectories that the planner collects to compute the eligibility reward.

- **Penalty**, which is the coefficient of reward penalty calculated the same as that in MOPO (Yu et al., 2020) and MAPLE (Chen et al., 2021).

- $H$, which is the policy rollout horizon.

- $E_{\text{model}}, E_{\text{policy}}, E_{\text{total}}$, which are the training epochs for model optimization, policy optimization, and outer loop respectively.

For ReDM-o experiments, there is an additional hyper-parameter $\alpha$ in Eq. 4. In the implementation, we actually multiply the coefficient on the loss terms defined by diversity and eligibility constraints, rather than the regularization term. Thus, we use $\alpha^*$ to denote the coefficient, and its value is listed in Table 3. For those parameters that are not involved in the tables, we kept them identical to MAPLE for a fair comparison.

## C.3  ENVIRONMENTS

**InvertedPendulum**. It is based on the CartPole enrironment. In this environment, there is a cart that can move linearly along a track. Attached to one end of the cart is a pole, which has another end that is free to move. The objective is to balance the pole on top of the cart by applying forces to the cart. The available actions in this environment involve pushing the cart either to the left or right.

**MountainCarContinuous**. The Mountain Car is a deterministic MDP that involves a car positioned stochastically at the bottom of a sinusoidal valley. In this version of the problem, the car can take continuous actions by applying accelerations in either direction. The objective of the MDP is to skillfully accelerate the car to reach the goal state situated on top of the right hill.

**Acrobot**. The Acrobot is a dynamic system comprised of two links connected linearly to form a chain, with one end of the chain fixed. The joint between the two links is actuated, allowing for the application of torques. The objective of the Acrobot task is to strategically apply torques to the actuated joint in order to swing the free end of the chain above a specified height, starting from an initial state where the chain hangs downwards.

Table 4: Performance of ReDM-o and other baselines on D4RL benchmark with limited data. All scores presented are averaged over 5 seeds and normalized by the way proposed by (Fu et al., 2020). We bold the highest mean. Hyper-parameters for each task can be found in Table 3.

| Env | Type | Size | ReDM-o | MAPLE | MOPO | CQL | IQL | TD3BC |
|---|---|---|---|---|---|---|---|---|
| HalfCheetah | Random | 200 | **2.5 ± 0.9** | 1.0 | 2.2 | −1.5 | 0.6 | −1.5 |
| HalfCheetah | Random | 5000 | **19.0 ± 1.5** | 9.3 | 2.4 | 1.7 | 2.2 | 9.5 |
| Hopper | Random | 200 | **23.8 ± 8.0** | 14.3 | 0.6 | 0.8 | 1.1 | 0.9 |
| Hopper | Random | 5000 | **31.4 ± 0.3** | 13.8 | 0.7 | 0.8 | 2.5 | 1.6 |

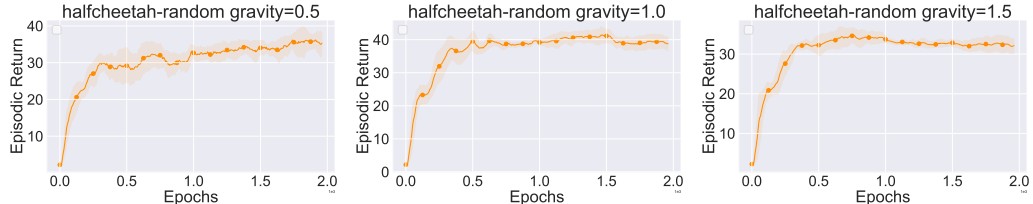

Figure 8: Learning curves of ReDM-o on D4RL halfcheetah-random dataset with different hyper-parameters. All results are averaged acroos 5 seeds.

**Hopper**. The Hopper environment in MuJoCo features a 2D one-legged robot, resembling a grasshopper or a pogo stick. The goal is to control the robot's actions to make it hop and maintain balance while moving forward. The agent must learn to apply appropriate forces to the leg joints to generate hopping motions and effectively navigate the environment.

**Walker2d**. The Walker2d environment simulates a 2D bipedal robot with two legs. The objective is to control the robot's movements to make it walk and maintain stability. The agent needs to learn how to generate coordinated leg movements and adjust joint angles to ensure smooth and balanced walking gaits.

**HalfCheetah**. The HalfCheetah environment represents a 2D model of a cheetah-like quadruped robot. The task is to control the robot's actions to achieve fast and efficient running. The agent must learn to generate coordinated leg movements and apply appropriate forces to the leg joints to maximize speed and maintain stability during locomotion.

# D    ADDITIONAL RESULTS OF EXPERIMENTS WITH LIMITED DATA

The detailed results of all the methods are shown in Table 4.

# E    ADDITIONAL RESULTS OF EXPERIMENTS WITH MISMATCHED DATA

We chose 12 tasks of D4RL, including 3 domains (*HalfCheetah*, *Hopper* and *Walker2D*) and 4 dataset qualities (*random*, *medium*, *medium-replay* and *medium-expert*). To create a mismatch setting, the test environment is modified by scaling its gravity by 0.5, 1.0 and 1.5, and 1.0 means the unbiased case. All results are listed in Table 5. ReDM-o outperforms other methods on the majority of the tasks

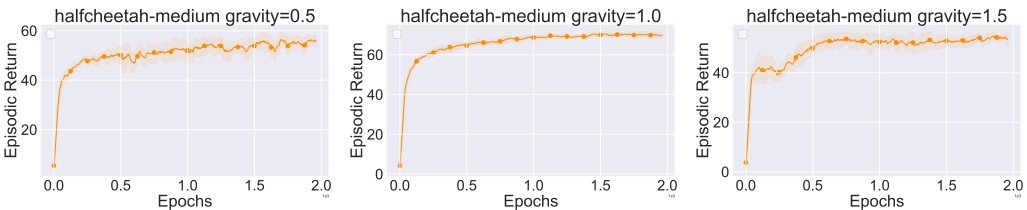

Figure 9: Learning curves of ReDM-o on D4RL halfcheetah-medium dataset with different hyper-parameters. All results are averaged acroos 5 seeds.

Table 5: Performance of ReDM-o and other baselines on D4RL benchmark. All scores presented are averaged over 5 seeds and normalized by the way proposed by (Fu et al., 2020). We bold the highest mean. Hyper-parameters for each task can be found in Table 3.

| Environment | Type | Parameter | ReDM-o | MAPLE | MOPO | IQL | TD3BC | CQL |
|---|---|---|---|---|---|---|---|---|
| HalfCheetah | random | Gravity-0.5 | **33.9 ± 3.9** | 32.2 | 32.2 | 11.0 | 11.0 | 12.0 |
| HalfCheetah | random | Gravity-1.0 | **39.7 ± 2.4** | 33.4 | 37.0 | 12.0 | 10.9 | 15.8 |
| HalfCheetah | random | Gravity-1.5 | **32.7 ± 1.5** | 29.7 | 32.4 | 11.6 | 8.2 | 10.2 |
| HalfCheetah | medium | Gravity-0.5 | 58.9 ± 2.7 | 58.1 | **60.1** | 39.5 | 43.7 | 41.3 |
| HalfCheetah | medium | Gravity-1.0 | 69.3 ± 0.2 | 70.1 | **73.5** | 48.3 | 48.1 | 47.0 |
| HalfCheetah | medium | Gravity-1.5 | **56.3 ± 1.8** | 53.2 | 54.2 | 40.0 | 38.4 | 37.8 |
| HalfCheetah | medium-replay | Gravity-0.5 | 56.2 ± 4.6 | **65.4** | 51.9 | 27.8 | 33.1 | 34.1 |
| HalfCheetah | medium-replay | Gravity-1.0 | 63.8 ± 1.7 | **65.6** | 47.0 | 43.9 | 44.7 | 45.2 |
| HalfCheetah | medium-replay | Gravity-1.5 | **52.1 ± 3.3** | 46.6 | 35.2 | 35.2 | 34.9 | 33.6 |
| HalfCheetah | medium-expert | Gravity-0.5 | **65.7 ± 2.6** | 63.4 | 42.1 | 39.1 | 47.6 | 50.5 |
| HalfCheetah | medium-expert | Gravity-1.0 | **96.1 ± 1.0** | 95.7 | 70.3 | 91.5 | 89.0 | 90.1 |
| HalfCheetah | medium-expert | Gravity-1.5 | **54.6 ± 1.6** | 52.7 | 47.4 | 32.2 | 36.3 | 50.1 |
| Walker2d | random | Gravity-0.5 | **21.6 ± 0.3** | 17.7 | 5.8 | 6.2 | 1.0 | 3.7 |
| Walker2d | random | Gravity-1.0 | **21.8 ± 0.2** | 21.8 | 5.1 | 4.3 | 1.0 | 3.6 |
| Walker2d | random | Gravity-1.5 | **21.7 ± 0.2** | 17.6 | 4.9 | 3.9 | 0.8 | 3.6 |
| Walker2d | medium | Gravity-0.5 | 37.2 ± 24.1 | 48.5 | **53.8** | 42.6 | 49.6 | 33.9 |
| Walker2d | medium | Gravity-1.0 | 78.4 ± 1.6 | 56.4 | 68.4 | 81.7 | **82.6** | 80.4 |
| Walker2d | medium | Gravity-1.5 | **52.4 ± 22.5** | 41.1 | 15.5 | 11.7 | 18.3 | 13.9 |
| Walker2d | medium-replay | Gravity-0.5 | 56.1 ± 2.8 | **67.4** | 56.2 | 52.8 | 49.3 | 37.7 |
| Walker2d | medium-replay | Gravity-1.0 | 70.1 ± 3.8 | 75.2 | **84.1** | 81.7 | 76.3 | 81.0 |
| Walker2d | medium-replay | Gravity-1.5 | **74.3 ± 2.8** | 42.4 | 34.6 | 32.0 | 14.7 | 18.0 |
| Walker2d | medium-expert | Gravity-0.5 | 74.1 ± 6.7 | **87.5** | 51.2 | 64.4 | 56.3 | 46.3 |
| Walker2d | medium-expert | Gravity-1.0 | 101.2 ± 2.3 | 109.1 | 43.6 | **112.5** | 110.3 | 107.1 |
| Walker2d | medium-expert | Gravity-1.5 | **82.6 ± 15.4** | 43.1 | 14.7 | 10.5 | 10.0 | 31.1 |
| Hopper | random | Gravity-0.5 | **31.5 ± 0.2** | 29.8 | 11.6 | 8.1 | 9.3 | 6.4 |
| Hopper | random | Gravity-1.0 | 31.6 ± 0.2 | **31.8** | 10.0 | 7.5 | 8.2 | 7.0 |
| Hopper | random | Gravity-1.5 | 31.5 ± 0.2 | **31.6** | 9.5 | 7.5 | 7.7 | 6.9 |
| Hopper | medium | Gravity-0.5 | **39.8 ± 28.6** | 36.7 | 11.0 | 13.4 | 13.0 | 26.6 |
| Hopper | medium | Gravity-1.0 | **104.7 ± 1.2** | 41.4 | 54.3 | 63.2 | 58.6 | 57.6 |
| Hopper | medium | Gravity-1.5 | **61.4 ± 38.2** | 38.9 | 33.8 | 25.0 | 15.6 | 18.1 |
| Hopper | medium-replay | Gravity-0.5 | 46.7 ± 26.4 | 55.3 | 33.1 | 12.5 | 21.3 | **93.0** |
| Hopper | medium-replay | Gravity-1.0 | **102.5 ± 0.8** | 58.4 | 100.9 | 90.6 | 58.4 | 55.0 |
| Hopper | medium-replay | Gravity-1.5 | **77.7 ± 31.9** | 23.5 | 22.3 | 21.5 | 14.6 | 18.0 |
| Hopper | medium-expert | Gravity-0.5 | 48.4 ± 31.0 | **59.6** | 12.8 | 21.6 | 17.1 | 46.3 |
| Hopper | medium-expert | Gravity-1.0 | **111.5 ± 0.8** | 64.2 | 34.0 | 87.9 | 103.0 | 109.7 |
| Hopper | medium-expert | Gravity-1.5 | 41.4 ± 34.7 | **56.3** | 10.7 | 14.5 | 23.7 | 31.1 |
| Average | - | Mismatched | **50.4** | 45.8 | 30.7 | 24.4 | 24.0 | 29.3 |
| Average | - | Matched | **74.2** | 60.3 | 52.4 | 60.4 | 57.6 | 58.3 |

(22/36), especially on mismatched tasks (17/24), which validates the generalization capability of ReDM-o. Additionally, we provide a demonstration of the average performance in different scenarios, specifically the *matched* scenario, which averages performances over all tasks with Gravity-1.0, and the *mismatched* scenario, which averages performances over all tasks with Gravity-0.5 and Gravity-1.5. These results highlight that our method performs well on both matched and mismatched tasks.

We also provide the learning curves of ReDM-o to illustrate the stability of our method. We take *halfcheetah-random* in Figure 8 and *halfcheetah-medium* in Figure 9 dataset as an example. The stable improvement in our method is evident from the curves obtained under three different values of gravity. We also find that in *halfcheetah-random*, since we choose 50 as the interval of generating new candidate dynamics models, an apparent improvement in the performance can be witnessed nearly every 50 epochs.

We conducted additional experiments on the standard D4RL Adroit tasks, which feature higher dimensions compared to previous tasks. For instance, the state dimension in the hammer tasks is 46, while in the pen tasks it is 45. The results, as presented in Table 6, demonstrate the effectiveness of our method in handling more complex tasks.

# F   ADDITIONAL RESULTS OF EXPERIMENTS WITHOUT INTERACTION DATA

We additionally provide the expert policy performance on InvertedPendulum, MountainCar and Acrobot, to demonstrate the extent of improvement ReDM achieved over random policies. We also

Table 6: Performance of ReDM-o and other baselines on D4RL Adroit benchmark with full dataset. All scores presented are averaged over 5 seeds and normalized by the way proposed by (Fu et al., 2020). We bold the highest mean. Hyper-parameters for each task can be found in Table 3.

| Environment | Type | ReDM-o | MAPLE | MOPO | CQL | TD3BC | IQL |
|---|---|---|---|---|---|---|---|
| Pen | Cloned | **57.3 ± 16.5** | 45.7 | 54.6 | 27.2 | −2.1 | 54.8 |
| Pen | Human | 35.7 ± 17.2 | 27.5 | 10.7 | 35.2 | −1.0 | **65.4** |
| Hammer | Cloned | **1.5 ± 0.5** | 0.9 | 0.5 | 1.4 | −0.1 | 1.1 |
| Hammer | Human | 0.3 ± 0.1 | 0.2 | 0.3 | 0.6 | 0.2 | **1.3** |



Figure 10: Relative performance between expert policy and the random policy on different environments with different hyper-parameters.

organize the results in the relative performance of the expert policy versus the random policy, and the results are illustrated in Figure 10. It is worthwhile to note that, although the expert policy can do better, ReDM managed to achieve about 30% and 70% of the expert performance in MountainCar and Acrobot respectively.

We also plot the learning curve of the meta-policy w.r.t. the number of generated candidate dynamics models, as shown in Figure 11. As the number of gradient steps increases, candidate dynamics are generated and included in the candidate set, resulting in gradual performance improvement of ReDM over the random policy in most of the tasks.

Finally, we provide the t-SNE results of the model rollouts with a total of 3 and 5 models respectively. We collect the model rollouts with the same procedure as described in the main text. Results in Figures 12∼13 validate that models generated by ReDM are more diversified.

## G  DETAILED RELATED WORK

**Model-Based Reinforcement Learning.** In complex tasks, traditional RL methods often require millions of interactions with the environment to fully optimize the policy (Haarnoja et al., 2018). In order to mitigate the issue of optimization, model-based reinforcement learning provides a promising approach by extracting a dynamics model from the interaction data to serve as the proxy environment (Janner et al., 2019; Luo et al., 2019) or provide extra generalization ability (Young et al., 2023; Ying et al., 2023; Lee et al., 2020). The most frequently used formulation of dynamics model learning is to maximize the one-step log-likelihood of the observed state transition (Janner et al.,

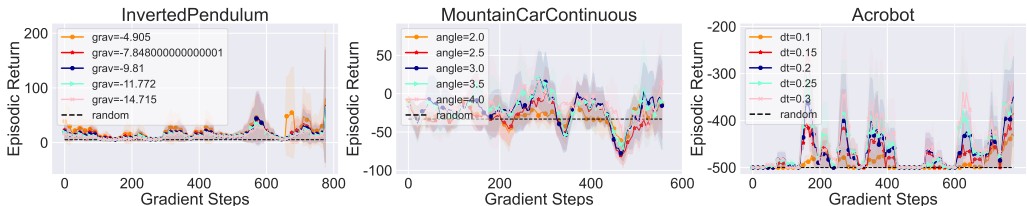

Figure 11: Learning curves of ReDM on different environments with different hyper-parameters. All results are averaged across 5 seeds.

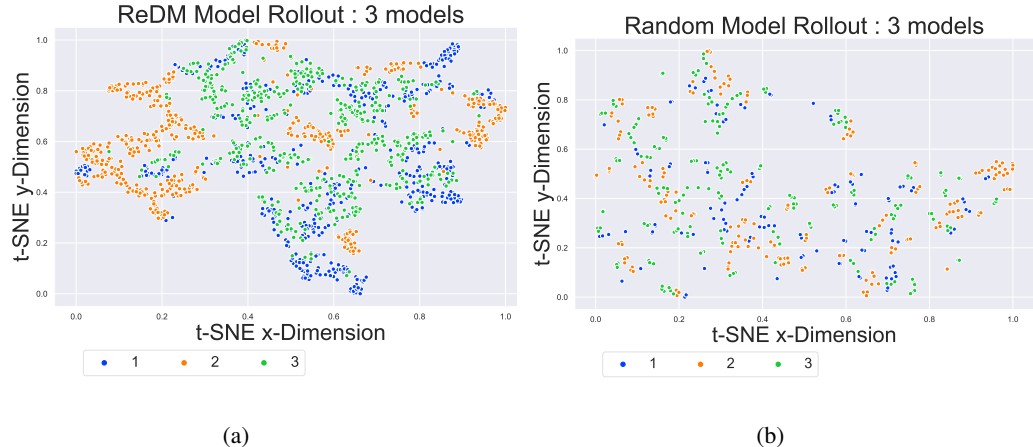

(a)
(b)

Figure 12: The t-SNE results of the data rollout from distinguished 3 generated models. (a) is the data from our generated models with no interaction data, and (b) is the data from random parameterized models.

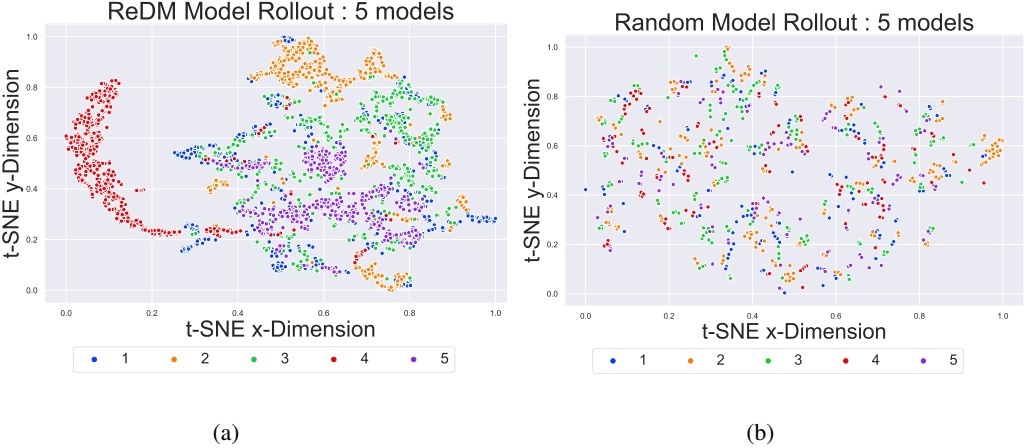

(a)
(b)

Figure 13: The t-SNE results of the data rollout from distinguished 5 generated models. (a) is the data from our generated models with no interaction data, and (b) is the data from random parameterized models.

2019; Luo et al., 2019; Clavera et al., 2018). However, such MLE-estimated dynamics models are known to suffer from compound error, which means the error in predicted states grows quadratically with the rollout horizon $H$. To control the compound error, MBPO (Janner et al., 2019) introduces branch rollout, which rollouts the policy for a short planning horizon starting from states chosen from the off-policy buffer uniformly. An alternative for model learning is to imitate the environments with GAIL (Ho & Ermon, 2016), whose error is proved to grow linearly with the horizon $H$ (Xu et al., 2020a).

Learning dynamics models can also benefit offline reinforcement learning in that they can augment the offline dataset with synthetic rollouts. Nevertheless, the inaccuracy of the learned models bears a more serious effect on policy optimization in the offline setting, as no more interaction data is available to provide the corrective feedback. To remedy this, a wide range of literature focuses on how to prevent the policy from accessing the uncertain area by modifying the learned MDP. For example, MOPO (Yu et al., 2020) and MOReL (Kidambi et al., 2020) learn an ensemble of models and utilize their disagreement to modify the model. RAMBO (Rigter et al., 2022), on the other hand, formulates the model learning as a zero-sum game with the policy and thus adversarially encourages the model to transition to low-value states. However, the above-mentioned methods all employ the principle of pessimism in the face of uncertainty, often leading to an over-conservative policy. A recent approach, MAPLE (Chen et al., 2021), learns a large set of models through supervised learning on the offline dataset and then learns a dynamics-aware policy that is capable of adapting to the real dynamics by identifying the most consistent dynamics model from the training ensemble. Although MAPLE is similar to our proposed method, the ensemble set of dynamics lacks diversity and eligibility, which hinders the overall performance of the final policy.

**Adaptive Policy Learning.** In order to adapt to different tasks or dynamics models, it is common to augment RL policies with a context encoder so that the policy can identify the environment during execution and make corresponding adjustments to its decision. The context encoder can be implemented with various architectures. A common approach is to use RNN-based networks to encode the up-to-date history and take RNN's final state as the context, as demonstrated by $RL^2$ and MAPLE (Chen et al., 2021). PEARL (Rakelly et al., 2019) extracts the context information via the multiplication of Gaussian distributions, each of which encodes the information carried by a single transition tuple obtained from the environment. On the other hand, recent studies have shown that the attention mechanism has an unparalleled ability in terms of feature extraction, as witnessed in natural language processing (Vaswani et al., 2017). Thus, some recent studies (Yang et al., 2020; Parisotto et al., 2020; Melo, 2022; Lin et al., 2022) adopted Transformer-like structures to encode features of transition tuples or trajectories. In this paper, we employ standard architectures of context encoders (RNN networks or Transformer encoders) to ensure a fair comparison against other baseline methods as our main focus is on how to efficiently generate a dynamics model.

**Unsupervised Reinforcement Learning.** Through the lens of unsupervised reinforcement learning, ReDM can be categorized as learning an effect policy without knowledge about the task dynamics. Previous works in unsupervised RL typically assume that the agent can interact with the online environment without the reward signal, or that it can access offline datasets without reward annotations. One line of research in unsupervised RL focuses on extracting meaningful skills that can later be utilized or composed to establish an effective policy when a task reward is given in the future. To realize this, they often seek to enhance the diversity among the policy population or the extracted skills (Eysenbach et al., 2018; Laskin et al., 2022; Parker-Holder et al., 2020; Sharma et al., 2019). Another strategy involves pre-training the agent to encapsulate environmental dynamics in representations, thereby enabling rapid adaptation to tasks upon the availability of rewards, as investigated by (Ghosh et al., 2023; Touati & Ollivier, 2021). In this paper, we examine a unique scenario where the reward function is known, but the model of dynamics remains undiscovered. Our method, ReDM, aligns with the broader unsupervised RL literature in its emphasis on diversifying the candidate dynamics set.

**Rehearsal Learning**. The concept of rehearsal is widely used in many areas of machine learning. In the context of causality (Zhou, 2022), the concept of rehearsal involves taking proactive measures to control actionable factors that can influence the occurrence or non-occurrence of desired or undesired events. By manipulating these factors, the goal is to increase the probability of the desired event and decrease the probability of the undesired event. In the context of continual learning (Liu et al., 2023; Yoon et al., 2022; Pelosin & Torsello, 2022), rehearsal is a technique used to mitigate catastrophic forgetting, which refers to the phenomenon where a model's performance on previously learned

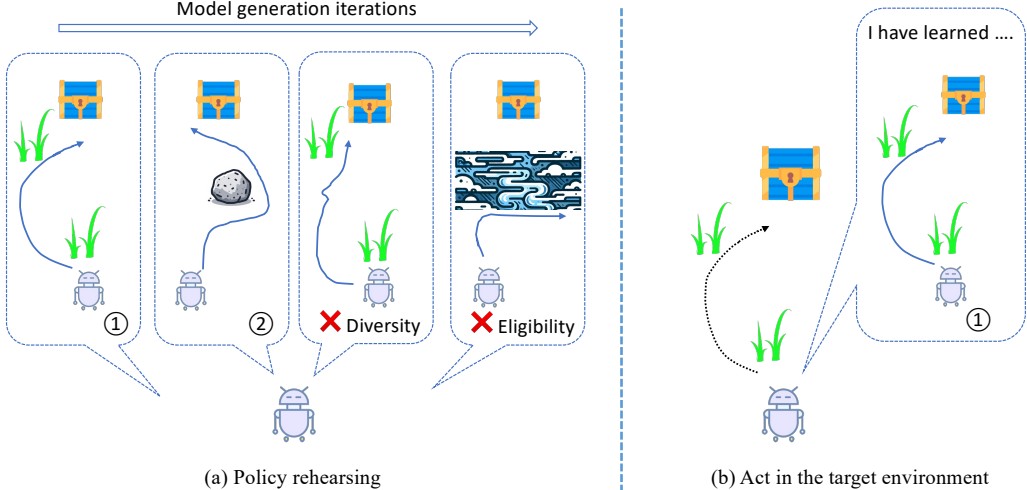

(a) Policy rehearsing        (b) Act in the target environment

Figure 14: An illustrative example of policy rehearsing when a robot try to find the treasure. During (a) policy rehearsing, the agent is trained through cycles of generating diverse and eligible candidate dynamics models. When (b) acting in the target environment, the agent subsequently adapts to the target environment by matching it to similar candidate models it have learned.

tasks degrades significantly when learning new tasks. The goal of rehearsal is to retain knowledge of previous tasks while learning new ones, allowing the model to maintain its performance across multiple tasks over time. Rehearsal involves storing and periodically revisiting a subset of previously encountered data samples (Jiang et al., 2023) or knowledge (Churamani et al., 2023) during the learning process. When training on new tasks, these stored samples are mixed with the current task's data to create a combined training set. By including past data, the model is exposed to a mixture of old and new information, which helps in preserving the knowledge acquired from previous tasks. There are also some other areas that involve rehearsal, such as nature language process (Araujo et al., 2023) and incremental learning (Jiang et al., 2023).

# H  AN ILLUSTRATION OF POLICY REHEARSING

In Figure 14, we demonstrate the idea of policy rehearsing and how it helps decision-making in the target environment. Supposing a task where a robot try to find treasures in the environment, the robot may envision possible routes and outcomes of accomplishing the task. This process reflects the dynamics model generation of ReDM. During the iterations, dynamics similar to previous outcomes or unable to complete the task will be masked out according to designed diversified and eligible metrics. A meta-policy is trained to adapt to those generated dynamics models and make decisions. When deployed to the target environment, the policy can identify it and build connections with previously generated dynamics. As a result, the agent can successfully adapt to the target environment with the assistance of policy rehearsing.

