# OpenReview forum: "Policy Rehearsing: Training Generalizable Policies for Reinforcement Learning"
_ICLR.cc/2024/Conference — ICLR 2024 poster_

### Official Review · Reviewer_AWzJ · 2023-10-20

**Soundness:** 3 good
**Presentation:** 2 fair
**Contribution:** 3 good
**Rating:** 6
**Confidence:** 4

**Summary:**

This work considers training an agent without online interaction or abundant offline data but only with the reward function of the target environment. Borrowing the idea of rehearsal from the cognitive mechanism, this work proposes policy rehearsal. In detail, this work hopes to train an array of models to imitate the target model. Theoretical analyses indicate that the target environment performance gap between the policy trained in these imitated models and the optimal policy can be bounded by three terms, which are further summarized as diversity and eligibility. Based on these two criteria, this work proposes two corresponding reward functions for training imitated models and then uses these models to train the policy. Also, the proposed ReDM can easily combined with offline datasets. Extensive results show the effectiveness of ReDM.

**Strengths:**

- The ideas about the setting are novel and important, minimizing interaction with the environment as much as possible is an important problem in the RL community. Also, introducing rehearsal into RL is novel and enlightening.

- The writing of Sec 3.2 is clear and solid, I have roughly read all the proofs, which are written quite clearly.

- The proposed ReDM utilizes two novel terms for learning an imitated model, which is interesting and helpful.

Currently, my evaluation of this paper is really Boardline. If authors can address my concerns in Weaknesses and Questions, or point out what I have misunderstood, I'd like to update my scores accordingly. Also, I will keep active in the following discussion stage.

**Weaknesses:**

- The connection between diversity and controlling $\epsilon_e, \epsilon_a$ is unclear. For example, if all environments are the same, i.e., there is no diversity, it is obvious that $\epsilon_a=0$ is minimal. There also needs more explanation about why $\epsilon_e$ can be controlled via diversity.

- Based on the previous points, one of my major concerns is why the proposed methods can help optimize the gap calculated in Thm 3.3. The authors have summarized the three errors in Thm 3.3 as diversity and eligibility, which indeed provides insights for analyzing this problem. But I think a more direct connection, like whether the objective in Sec 3.3 can be proven to directly control the three errors in Thm 3.3, will make the analyses more solid.

- In experiments, providing the results directly trained in the target environments as the reference will better show the results.

- Lack of some related works, like utilizing model-based methods for improving generalization [1-3], and finding diverse skills for unsupervised RL [4-6] as this work hopes to find diverse models.

[1] Context-aware Dynamics Model for Generalization in Model-Based Reinforcement Learning

[2] Task Aware Dreamer for Task Generalization in Reinforcement Learning

[3] The Benefits of Model-Based Generalization in Reinforcement Learning

[4] Diversity is All You Need: Learning Skills without a Reward Function

[5] Effective diversity in population based reinforcement learning

**Questions:**

- In my opinion, the considered setting is that the agent can only get the reward function of the target task but has no knowledge about the dynamic of the target task. Is it right? Given the offline data, it is understandable that the agent can learn the dynamic to some degree. But without an offline dataset, it seems that there is no idea for the agent to learn the dynamic of the target task.

- Based on the previous question, I'm confused about the setting of Experiment 4.1 " ReDM With no Interaction Data". As there are no data about the environment and the agent can not interact with the environment, how does the agent to learn about the environment?

- As Unsupervised RL considers training an agent in the environment without reward, in my opinion, the setting in this work is like training an agent and models in the environment with reward but without dynamic. As the dynamic of the target environment will vary a lot, whether finetuning the agent (as well as the model) in the target environment with few steps will be more reasonable?

- About $r_e$ for Eligibility. The proposed method is to randomly sample N trajectories and estimate the biggest return. Is this inefficient as the state space and action space are continuous in experiments? Also, what is the choice of N in experiments?

- I'm curious about the performance of ReDM in the D4RL setting (Sec. 4.3) but without any Interaction Data.

---

> ### Author Response · Authors · 2023-11-19
> **Author response (Part 1/2)**
>
> Thank you for your interest in our paper, we briefly summarized your concerns and questions as follows:
>
> **Comment1**: The connection between diversity and controlling \$\\epsilon_a\$ and \$\\epsilon_e\$ is unclear.
>
> **A1**: We did not claim that diversity can control the error term \$\\epsilon_e\$ in the paper, so we conjecture that you were actually questioning the relationship between \$\\epsilon_a, \\epsilon_m\$ and diversity. To control the error term \$\\epsilon_m\$, the candidate models should be dispersed throughout the hypothesis space of the dynamics. This ensures that the target environment falls within the vicinity of one of the candidate dynamics with a high likelihood. Conversely, if the candidate dynamics are concentrated within a limited area, the target environment risks being an outlier.
>
> As for \$\\epsilon_a\$, we conjecture that with a diversified candidate model, it should be easy for the meta-policy to differentiate the context. However, this is an intuition without rigorous proof, and we have removed this statement.
>
> **Comment2**: Can you prove that the objectives can control the error terms?
>
> **A2**: The error term \$\\epsilon_e\$ requires the candidate dynamics to contain paths toward high rewards. We formulate this as a reinforcement learning task, by treating the dynamics as an agent and optimizing for the reward achieved by a fixed planner. Regarding \$\\epsilon_m\$, after building the connection between controlling this term and enhancing the diversity, we adopted an adversarial methodology which is proven to generate distinguished candidate models (Lemma 3.4). we will continue to explore how to optimize these errors more directly in future work.
>
> **Comment3**: In experiments, providing the results directly trained in the target environments as the reference will better show the results.
>
> **A3**: We trained expert policies directly in the three environments using SAC, and the relative performances (the performance of SAC minus that of the random policy) are listed in the following table. Although expert policies achieve higher scores, we find that ReDM still manages to obtain about 30% and 70% of the expert’s performances in MountainCar and Acrobot respectively. We think this is notable considering that ReDM requires no interaction with the environment. More details about this can be found in Appendix F.
>
> | Environment | grav=0.5 | grav=0.8 | grav=1.0 | grav=1.2 | grav=1.5 |
> | -- | -- | -- | -- | -- | -- |
> | InvertedPendulum | 993.5 | 991.8 | 994.8 | 992.5 | 993.8 |
>
> | Environment | angle=2.0 | angle=2.5 | angle=3.0 | angle=3.5 | grav=4.0 |
> | -- | -- | -- | -- | -- | -- |
> | MountainCarContinuous | 128.7 | 128.4 | 128.5 | 128.4 | 106.8 |
>
> | Environment | dt=0.10 | dt=0.15 | dt=0.20 | dt=0.25 | dt=0.30 |
> | -- | -- | -- | -- | -- | -- |
> | Acrobot | 375.2 | 414.7 | 428.6 | 435.9 | 445.6 |
>
> **Comment4**: Related work on utilizing model-based methods for improving generalization and skill discovery in unsupervised RL is missing.
>
> **A4**: Thanks for pointing out this. We updated Section 2 in the main text to discuss about papers that are most related to ReDM. It's worth noting that a detailed discussion about related works is deferred to Appendix G due to space limits. We also updated this appendix section to include papers about generalization and unsupervised RL based on your recommendation.

---

> ### Author Response · Authors · 2023-11-19
> **Author response (Part 2/2)**
>
> **Q1**: Without an offline dataset, it seems that there is no idea for the agent to learn about the dynamics.
>
> **A1**: We would like to clarify that ReDM does NOT learn the target dynamics. ReDM generates diversified and eligible candidate models. ReDM then trains a meta-policy in these models, so that the meta-policy is able to generalize over different dynamics. Without any interaction data, the generated candidate models are expected to represent the whole model space. In such cases, the meta-policy can adapt to any environment including the real environment.
>
> **Q2**: Relation to unsupervised RL and potential of incorporating fine-tuning?
>
> **A2**: It is quite novel and interesting to interpret rehearsal as a kind of unsupervised RL without dynamics, and we have added a discussion about the relation in Appendix G. As for fine-tuning, it is worth noting that in Section 4.1 Figure 2, we conduct a fine-tuning experiment on InvertedPendulum, where we fine-tuned the learned meta-policy with online trajectories. Our meta-policy can achieve fast adaptation compared to directly learning from online trajectories.
>
> **Q3**: Is random sampling inefficient? What is the choice of N?
>
> **A3**: We agree that a random shooting planner can be less efficient and there are more advanced methods that can be applied instead. However, for the purpose of concept-proof and simplicity, we show that with such a simple method ReDM can also yield good results. N is set to \$50\$. We have updated the hyper-parameter table in Appendix C.2.
>
> **Q4**: The performance of ReDM in the D4RL setting but without any interaction data?
>
> **A4**: As the state dimensionality increases, ReDM requires much more computation power to generate sufficient many models for training an effective meta-policy. We will report the results on D4RL benchmarks. In these cases, we simultaneously explore ways to incorporate more constraints to reduce the model space, such as ReDM-o that can incorporates mismatching data.
>
> We hope the above explanation can answer your questions, but we are willing to follow up if you have further questions.

---

> > ### Comment · Reviewer_AWzJ · 2023-11-20
> > **Thanks for your reply**
> >
> > I appreciate the authors' reply and supplemented experiments and I have read other reviewers' comments also. Most of my concerns are addressed.
> >
> > - The connection between calculated loss terms and proposed methods is intuitive, not theoretically analyzed. And I believe further analyses can make it more solid.
> >
> > - I understand that "ReDM does not learn the target dynamics, and it generates diversified and eligible candidate models for training a meta-policy in these models". However, during the evaluation stage, if we no longer finetune the policy and hope the policy performs well. It seems that we can get a meta-policy that can handle any dynamics, which seems to be too difficult. As meta RL / unsupervised RL will train meta-policy for fast adapting to new tasks, I think more analyses and experiments considering fast adaption will be more reasonable.
> >
> > For all that, I actually believe that the proposed setting is novel and important, which is an important problem in the RL community. Thus I have raised my score to 6.

---

> > > ### Author Response · Authors · 2023-11-21
> > > **Author response**
> > >
> > > We are glad to address your concerns. We will continue to investigate these interesting problems in future work.

---

### Official Review · Reviewer_GFGi · 2023-10-30

**Soundness:** 4 excellent
**Presentation:** 2 fair
**Contribution:** 4 excellent
**Rating:** 8
**Confidence:** 3

**Summary:**

This paper presents a pretty interesting idea called rehearsal, which is able to **initialize or warm up a generalizable policy with zero interaction data or limited mismatched offline data**. Concretely, the proposed method, *ReDM*, takes as input a reward function and a termination function and generates a set of transition functions or models. Imaginary trajectories can thus be generated by rolling out these transition models and used to warm up the policy. As some of the models may produce data close to the target environment dynamics, the policy warmed up with these data can have a good initialization when deployed to the target environment, which is helpful for subsequent fine-tuning. Additionally, the method can be modified for offline-RL settings, allowing it to learn a robust and generalizable policy even with a small amount of offline data mismatched with target environment dynamics.

The method is motivated theoretically and contains lots of analysis like performance bound, laying foundations for future study in this new direction. Besides, the experiments on the standard gym and D4RL environment empirically prove the effectiveness of the method for both online and offline policy learning.

**Strengths:**

1. The idea is novel unlike traditional model-based RL, this new idea suggests learning a bunch of transition models from reward function and termination functions, exempting the need for interaction data.
2. In terms of soundness, it proves empirically and theoretically that the transition models learned in this way can help warm up the policy and improve its performance when deployed in environments with diverse transition dynamics.

**Weaknesses:**

1. The paper writing is not attractive. In my perspective, the main paper contains too much tedious content regarding the theoretical analysis and lacks an explanation for the rehearsal framework. My suggestion would be to move some theoretical content to the appendix and include at least one figure to explain the procedures of this new rehearsal framework and what it can achieve or why we need it. People don't care about the theoretical stuff until they are attracted by the idea and want to dive into it. Thus I suggest making some figures to explain the idea or the method.
2. No standard deviation is included for experiments in Table 1. Also, there is no error bar in Figure 7.
3. What is the $D_{TV}$ should be explained in the main paper. It is strongly related to your main theorem but without definition.
4. What is relative performance? Is it calculated through minus the baseline performance?
5. The axis *Number of models* in Figure 3 should be [0, 10, 20, 30, 40], right?

**Questions:**

1. How about replacing the random model for calculating the eligible reward with a human-crafted planner? It is supposed to be helpful for improving the performance as well. I guess this can be a good direction for exploration and to make this method more practical. A simple rule-based planner is also as easily accessible as a reward function in most practical settings like robotics.
2. In the zero interaction data setting, the method indeed works well in three simple gym environments. I wonder if the method still works well in the more complex Mujoco environment without any pre-collected interaction data. I am curious about its performance on high-dimensional control tasks.

---

> ### Author Response · Authors · 2023-11-19
> **Author response**
>
> We would like to express our gratitude for your constructive feedback on our paper. We have revised our paper based on your suggestions, including:
>
> - **Using Figure 15 in Appendix H to explain the idea of policy rehearsing.** We do not use the figure to replace the theory part in our main paper currently since there are concerns about the theories from other reviewers. However, we are willing to reorganize the paper structure once these concerns are addressed.
> - **Adding standard deviation for Table 1.** However, we did not add an error bar to Figure 7 since the results are aggregated over different datasets and gravity levels. Instead, we provide the mean and standard deviation of the performance for each task in Appendix E. Furthermore, we have revised our paper here to enhance clarity in our expression.
> - **Explaining \$D_\{TV\}\$ in Theorem 3.3.**
> - **Explaining the meaning of relative performance.** Yes, it is calculated by minus the baseline performance.
> - **Changing the ticks of the x-axis in Figure 3.**
>
> For the remaining questions:
>
> **Q1**: How about replacing the random shooting planner with a human-crafted planner?
>
> **A1**: Thanks for your suggestions. We agree that a random shooting planner can be less efficient and there are more advanced methods that can be applied instead. However, for the purpose of concept-proof and simplicity, we show that with such a simple method ReDM can also yield good results.
>
> **Q2**: ReDM's performance in complex MuJoCo environments without offline data.
>
> **A2**: As the state dimensionality increases, ReDM requires much more computation power to generate sufficient many models for training an effective meta-policy. We will report the results on D4RL benchmarks. In these cases, we simultaneously explore ways to incorporate more constraints to reduce the model space, such as ReDM-o that can incorporates mismatching data.
>
> We hope the explanation provided above addresses your questions. However, we are more than willing to provide further clarification or address any additional inquiries you may have.

---

> > ### Comment · Reviewer_GFGi · 2023-11-21
> > **Thank you**
> >
> > Thank you for the response. Most of my concerns are addressed.
> >
> > As a new idea, it is thoroughly evaluated with comprehensive experiments and theories. I would like to keep my score unchanged.

---

### Official Review · Reviewer_anZu · 2023-10-31

**Soundness:** 3 good
**Presentation:** 2 fair
**Contribution:** 3 good
**Rating:** 8
**Confidence:** 3

**Summary:**

The paper proposes a method for offline model-based reinforcement learning. The idea is to generate a set of candidate dynamics models and learn an adaptive policy that optimizes the original reward on this candidate set. If the true dynamics are in the distribution of the candidate set, the adaptive policy should perform well on the true task. The central problem lies in generating a candidate set of dynamics models. The authors propose optimizing over dynamics models with RL using a reward that incentivizes (1) diversity among the set and (2) the tendency for random trajectories to achieve high reward. The method alternates between optimizing for a new dynamics model to add to the set and optimizing for a new adaptive policy given the current set. When interaction data from the true task is available, it is used to regularize the optimization over dynamics models. Experiments show the method can work with no interaction data on low-dimensional continuous control tasks (inverted pendulum, mountain car, acrobot). On two D4RL tasks (hopper, half-cheetah) with a small amount of random interaction data, the method outperforms prior offline model-free and model-based RL methods.

**Strengths:**

The method is similar to MAPLE but replaces the dynamics model generation process with a more directed procedure (RL on a custom reward vs learning an ensemble of models). The reward used in the dynamics model generation process is well motivated by formal analysis of error bounds. The different components of the method are analyzed/ablated.

**Weaknesses:**

My main concern is the limited applicability of this method beyond low-dimensional benchmark tasks due to some significant assumptions. The method assumes access to a query-able reward/termination function and the initial state distribution. Though more importantly, the method assumes that the dynamics can be easily parameterized and optimized over with RL. Additionally, the method assumes that random plans through the candidate dynamics models will achieve some non-zero reward (to optimize the dynamics models for the eligibility reward). These assumptions makes the method difficult to apply (if not impossible) in sparse-reward or high-dimensional (e.g image-based) environments. In principle these issues could be solved by providing the method with enough interaction data to learn a good dynamics model initialization. However, then prior offline model-based or model-free methods might also work well. Additionally, this still wouldn't make the method applicable to sparse reward problems.

Another concern is the limited scope of the experiments relative to prior work. The evaluations on InvertedPendulum, MountainCar, and Acrobot are good for analyzing the method, however for the comparison to prior work, experiments are only shown for HalfCheetah and Hopper. It would be good to additionally include at least Walker2d. Additionally, the experiments with interaction data only test random interaction data and relatively small amounts of data (200 and 5000 transitions). While it is understandable that this is the setting where the proposed method would excel, it would be good to also show comparison to prior work with interaction data of varying optimality and amounts (including the full D4RL datsets).

There is no discussion of MAPLE in the related work section. MAPLE is very related (just a different model generation process) so the similarities and differences should be addressed here. It would also be good to include a brief mention of meta-learning in the related work as the proposed method uses similar concepts when optimizing for the adaptive policy.

Smaller comments:
- Algorithm 1 does not say a lot about the method. It could be replaced by algorithms 5/6 from the appendix.
- Figure 1 should use a more descriptive x-axis label like "Tasks".
- Figure 3 needs a more descriptive caption that explains what "model loss" means here.
- The locations of Figure 2 and 3 should be switched.

**Questions:**

- The explanation of the optimal policy gap is confusing. Specifically this sentence: "This discrepancy highlights the candidate model set’s capability to derive a proficient policy in the model itself." Does "model" here mean the true dynamics?
- "we conjecture that a diversified dynamics model set will correspond to a smaller ϵa since recognizing the dynamics is much easier" It's not clear to me why a more diverse candidate model would lower the adaptation cost. Could you explain this?
- In Figure 6, what is the shown performance relative to? Is this the performance of the policy at each iteration in the model at that iteration minus the performance of the policy at that iteration in the ground truth model?
- Figure 7: Are these results averaged over Hopper and HalfCheetah and averaged over each gravity level?

---

> ### Author Response · Authors · 2023-11-19
> **Author response (Part 1/2)**
>
> Thank you for your suggestions and comments. For those smaller comments you mentioned, we have fixed those issues and updated our paper. In the following, we provide more explanation for the rest questions and concerns.
>
> **Comment1**: Significant assumptions that may limit the applicability of ReDM: (1) random plans to achieve non-zero data; (2) query-able reward, initial state distribution and parameterized dynamics.
>
> **A1**: **About planners**: We agree that a random shooting planner can be less efficient and there are more advanced methods that can be applied instead. However, for the purpose of concept-proof and simplicity, we show that with such a simple method ReDM can also yield good results.
>
> **About reward, termination functions, and initial state distribution.** In many real-world applications of reinforcement learning, human experts are aware of the basic properties of the task, including the reward function, termination function, and initial states. These properties can often be cheaper to obtain than interaction data. Therefore, assuming known reward and termination functions are acceptable in these situations. Meanwhile, we will continue to explore more methods without these assumptions in the future.
>
> **About parameterization of dynamics models.** We employ neural networks to represent the dynamics, which can have strong expressive power and have been extensively applied in previous model-based RL studies (e.g., MBPO, MOPO, MuZero, etc). Thus we don't think this is an overly strong assumption.
>
> **Comment2**: Another concern is the limited scope of the experiments relative to prior work. It would be good to additionally include at least Walker2d. Additionally, it would be good to also show comparisons to prior work with interaction data of varying optimality and amounts.
>
> **A2**: We actually did the comparison between ReDM-o with other baseline offline RL methods using full D4RL datasets including Walker2D, and the aggregated results are presented in Section 4.3 Figure 7 in the manuscript. Detailed results are listed in Table 5 in Appendix E, where *Gravity-1.0* corresponds to standard offline RL tasks, and others mean the dataset and the target environment are mismatched. We also updated our paper and included some D4RL Adroit tasks in the following table, which also shows the effectiveness of our method.
>
> | Environment | Type | ReDM-o | MAPLE | MOPO | CQL | TD3BC |
> | --- | --- | --- | --- | --- | --- | --- |
> | Pen | Cloned | 57.3 $\\pm$ 16.5 | 45.7 | 54.6 | 27.2 | -2.1 |
> | Pen | Human | 35.7 $\\pm$ 17.2 | 27.5 | 10.7 | 35.2 | -1.0 |
> | Hammer | Cloned | 1.5 $\\pm$ 0.5 | 0.9 | 0.5 | 1.4 | -0.1 |
> | Hammer | Human | 0.3 $\\pm$ 0.1 | 0.2 | 0.3 | 0.6 | 0.2 |
>
> **Comment3**: MAPLE is very related and deserves discussion in the related work.
>
> **A3**: Thank you for your suggestions. We discussed with MAPLE in Appendix G, and we have revised to move the discussion into Related Work. We would like to clarify that MAPLE does not focus on model generation, and its models are learned from data with some randomness. ReDM studies how to generate models for training generalizable policies, and shows to work even with zero data or with mismatching data.

---

> ### Author Response · Authors · 2023-11-19
> **Author response (Part 2/2)**
>
> **Q1**: The explanation of the optimal policy gap is confusing. Specifically this sentence: "This discrepancy highlights the candidate model set’s capability to derive a proficient policy in the model itself." Does "model" here mean the true dynamics?
>
> **A1**: Sorry for the confusing expression. The "model" refers to the learned candidate model for policy optimization instead of the real dynamics. The *optimal policy gap* refers to the difference in performance between the optimal policy in the true dynamics and the best policy found in each candidate model. A large optimal policy gap indicates that the learned model has a large difference from the real dynamics.
>
> **Q2**: It's not clear why a more diverse candidate model would lower the adaptation cost.
>
> **A2**: This statement is only an intuition without proof. We will revise and remove it.
>
> **Q3**: In Figure 6, what is the shown performance relative to? Is this the performance of the policy at each iteration in the model at that iteration minus the performance of the policy at that iteration in the ground truth model?
>
> **A3**: The relative performance at iteration $i$ is calculated as (return of the final policy in the $i$-th model) - (return of the final policy in the ground truth model). We will revise to make this clear.
>
> **Q4**: Figure 7: Are these results averaged over Hopper and HalfCheetah and averaged over each gravity level?
>
> **A4**: Yes, the results are averaged over Halfcheetah, Hopper and **Walker2D** and over all gravity levels and datasets in D4RL MuJoCo. The detailed result for each domain, dataset and gravity level is listed in Table 5 in Appendix E.
>
> We hope our explanation and clarification can address your concerns. We are willing to offer further clarification or address any additional questions you may have.

---

> > ### Comment · Reviewer_anZu · 2023-11-21
> >
> > Thank you for answering my questions. The additional results on Adroit are helpful for further evaluating the performance of this method relative to prior work. Could you also add the results from the IQL paper to the Adroit results table? This would make it consistent with the other results and I think IQL may outperform ReDM-o on a few of these tasks.
> >
> > For the D4RL experiments, could you show the average performance for each gravity setting? It would be interesting to see if ReDM-o still outperforms the other methods when the gravity in the evaluation environment matches the gravity in the offline data.
> >
> > In Figure 3, does "fine-tuning" the adaptive policy just mean conditioning the adaptive policy on a few episodes of context? Or does it mean actual fine-tuning of the policy's parameters? I would expect that it means giving the adaptive policy context episodes, but this should be clarified.
> >
> > It looks like the revised appendix contains some implementation details of ReDM-o that seem important enough to be in the main paper. Specifically, it says that the method starts from a pre-trained policy and that the pre-trained policy replaces the planner. Additionally, the appendix discusses a "reward penalty" hyper-parameter that is not mentioned in the main paper. What does this reward penalty do?

---

> > > ### Author Response · Authors · 2023-11-21
> > > **Author response**
> > >
> > > Thank you for your suggestions. For the comments you mentioned, we provide more results and clarifications and revise our paper.
> > >
> > > **Add results from the IQL paper to Adroit results table**: We have updated our table by including the results of IQL. While the original paper presented the results of Adroit in the v0 version, we now report the results of IQL on the v1 version. It is worth noting that IQL demonstrates strong performance on several tasks. However, our method continues to maintain its effectiveness, showcasing competitive results across the evaluated tasks.
> > >
> > > | Environment | Type | ReDM-o | MAPLE | MOPO | CQL | TD3BC | IQL(paper v0)| IQL |
> > > | --- | --- | --- | --- | --- | --- | --- | --- | --- |
> > > | Pen | Cloned | 57.3 $\\pm$ 16.5 | 45.7 | 54.6 | 27.2 | -2.1 | 37.3      | 54.8 |
> > > | Pen | Human | 35.7 $\\pm$ 17.2 | 27.5 | 10.7 | 35.2 | -1.0 | 71.5       | 65.4 |
> > > | Hammer | Cloned | 1.5 $\\pm$ 0.5 | 0.9 | 0.5 | 1.4 | -0.1 | 2.1         | 1.1  |
> > > | Hammer | Human | 0.3 $\\pm$ 0.1 | 0.2 | 0.3 | 0.6 | 0.2 | 1.4           | 1.3  |
> > >
> > > **Show the average performance for each gravity setting**: Thank you for your suggestion. We separately calculate the average performance of the *matched tasks*, which averages performances over all tasks with Gravity-1.0, and that of *mismatched* tasks, which averages performances over all tasks with Gravity-0.5 and Gravity-1.5. The results still show that our method can perform well on both matched and mismatched tasks.
> > >
> > > **Does "fine-tuning" the adaptive policy just mean conditioning the adaptive policy on a few episodes of context?**: We utilize the parameters of the learned policy as an initialization. Subsequently, we fine-tune this initialized policy with a few online trajectory collected by the policy, and then we compare its performance to learning from scratch with the same policy architecture and the same amount of data.
> > >
> > > **What does this reward penalty do?**: The reward penalty technique is commonly used in model-based methods for balancing the degree of conservatism of the policy, and we employ the same calculation as MOPO[1] and MAPLE[2] for this term.
> > >
> > > [1] Yu et al. MOPO: Model-based Offline Policy Optimization.
> > >
> > > [2] Chen et al. Offline Model-based Adaptable Policy Learning.

---

> > > > ### Comment · Reviewer_anZu · 2023-11-21
> > > >
> > > > Thank you for answering my questions. While I don't believe this method would outperform prior work in high-dimensional settings, I still think it will be of interest to the community for settings where there is a small amount of data or the data is mismatched/sub-optimal. It's particularly interesting that this method can work in some settings even with no interaction data by using knowledge of the reward function, termination function, and initial state distribution. Thus, I have raised my score to an 8.

---

### Official Review · Reviewer_YzNF · 2023-11-01

**Soundness:** 2 fair
**Presentation:** 3 good
**Contribution:** 3 good
**Rating:** 8
**Confidence:** 3

**Summary:**

Human beings can make adaptive decisions in a preparatory manner, i.e., by making preparations in advance, which offers significant advantages in scenarios where both online and offline experiences are expensive and limited. Meanwhile, current reinforcement learning methods commonly rely on numerous environment interactions but hardly obtain generalizable policies. In this paper, the authors introduce the idea of *rehearsal* into policy optimization, where the agent plans for all possible outcomes in mind and acts adaptively according to actual responses from the environment. To effectively rehearse, they propose ReDM, an algorithm that generates a diverse and eligible set of dynamics models and then rehearse the policy via adaptive training on the generated model set. Rehearsal enables the policy to make decision plans for various hypothetical dynamics and to natually generalize to previously unseen environments. Their experimental results demonstrate that ReDM is capable of learning a valid policy solely through rehearsal, even with zero interaction data. Besides, they further extend ReDM to scenarios where limited or mismatched interaction data is available. The provided empirical results reveal that ReDM produces high-performing policies compared with other offline RL baselines.

**Strengths:**

1. The problem of policy rehearsing in offline reinforcement learning is interesting and challenging as an academic topic.
2. The description to the problem modeling and the methods is clear and generally easy-understanding.
3. The proposed method is well motivated by comprehensive preliminary theoretical analysis.
4. The experiment analysis is in-depth and insightful, which helps the readers bettere understand the effectiveness and underlying mechanism of the propose methods.

**Weaknesses:**

1. The environments used in the experiments are still limited. I encourage to supplement more environments to demonstrate the applicability of your proposed method is possible. Otherwise, we may argue if the solution can only be effective on some specific kinds of tasks.
2. Considering the proposed method needs to train the new dynamics models and meta-policy simultaneously, the complexity of this method and the training stability/convegence are encouraged to be clarified and analyzed.
3. The assumed accessibility to the task reward function and initial state distribution is often unrealistic in the real applications.

**Questions:**

1. I am curious if totally no interaction data, how can the generated dynamics model approximates the real dynamics in the target environment. It seems there lacks enough grounding points to support this potential. Does there exist the probability that the generated dynamics models are far from the dynamics in the target environment? I hope to see more analysis on this during the rebuttal.
2. The D4RL benchmark in your experiments is all Mujoco tasks with low input dimensions. Could you please consider incorporating some more high-dimensional task, in which the hypothesis space is too large to narrow down?
3. In the paper, you claim that the interaction data is only used to narrow down the hypothesis space. But could you please consider how to utilize these interaction data in a more direct way to better facilitate the policy learning as the complement to the purely dynamics model learning, like finetuning the learned meta policy? Besides, I cannot agree the statement that the biasedness in the interaction data will somehow hinder the policy optimization in traditional offline RL methods. If such pre-collected trajectories are expert ones or near-optimal ones, such *biasedness* can actually help avoid some low-value and dangerous states.
4. Considering your method encourages the diversity in the model learning part, some learned dynamics models may be unreasonable though the meta policy can still achieve high returns via planning in such models, like violating the physics laws or economics laws. And I can hardly expect the *eligibility* part in your method can help alleviate this 'short-path' issue. More explanations and discussions are encouaged during the rebuttal phase.

---

> ### Author Response · Authors · 2023-11-19
> **Author response**
>
> Thanks for your constructive comments. We provide more clarifications and empirical results for your questions and concerns.
>
> **Comment 1**: The environments used in the experiment are limited and more environments should be included.
>
> **A1**: Thank you for your suggestions. We have conducted experiments on 6 environments and will include more environments in the revised version. For example, we are conducting more experiments on D4RL Adroit tasks. The results on 4 of the environments are listed below, showing that the proposed method is still effective.
>
> | Environment | Type | ReDM-o | MAPLE | MOPO | CQL | TD3BC |
> | --- | --- | --- | --- | --- | --- | --- |
> | Pen | Cloned | 57.3 $\\pm$ 16.5 | 45.7 | 54.6 | 27.2 | -2.1 |
> | Pen | Human | 35.7 $\\pm$ 17.2 | 27.5 | 10.7 | 35.2 | -1.0 |
> | Hammer | Cloned | 1.5 $\\pm$ 0.5 | 0.9 | 0.5 | 1.4 | -0.1 |
> | Hammer | Human | 0.3 $\\pm$ 0.1 | 0.2 | 0.3 | 0.6 | 0.2 |
>
> **Comment 2**: ... needs to train the new dynamics models and meta-policy simultaneously, ... the training stability/convergence are encouraged to be clarified and analyzed.
>
> **A2**: We would like to clarify that the dynamics models and meta-policy are trained alternately. Such alternating training has been widely employed in solving minimax problems. We will revise to discuss that when there are sufficiently many dynamics models generated, the models can cover the model space, and thus the performance of the meta-policy should converge.
>
> **Comment 3**: The assumption of access to task reward and initial state distribution is unrealistic.
>
> **A3**: **About reward, termination functions, and initial state distribution.** In many real-world applications of reinforcement learning, human experts are aware of the basic properties of the task, including the reward function, termination function, and initial states. These properties can often be cheaper to obtain than interaction data. Therefore, assuming known reward and termination functions are acceptable in these situations. Meanwhile, we will continue to explore more methods without these assumptions in the future.
>
> **About parameterization of dynamics models.** We employ neural networks to represent the dynamics, which can have strong expressive power and have been extensively applied in previous model-based RL studies (e.g., MBPO, MOPO, MuZero, etc). Thus we don't think this is an overly strong assumption.
>
> **Q1**: How can the generated dynamics model approximate the real dynamics without any interaction data? Does there exist the probability that the generated dynamics models are far from the dynamics in the target environment?
>
> **A1**: ReDM does not approximate the real dynamics as we explained in **Common Question 1**. ReDM generates diverse dynamics models and thus dynamics models far from the real dynamics are often generated. These diverse dynamics models help train strongly generalizable meta-policies. We also present ReDM-o for when some interaction data is available. ReDM-o can generate dynamics models around the real dynamics.
>
> **Q2**: The D4RL benchmark in your experiments is all Mujoco tasks with low input dimensions. Could you please consider incorporating some more high-dimensional tasks, in which the hypothesis space is too large to narrow down?
>
> **A2**: Please refer to our response to **Comment 1**.
>
> **Q3**: I cannot agree with the statement that the biasedness of the interaction data will hinder policy optimization in traditional offline RL methods.
>
> **A3**: We would like to clarify that our purpose is to obtain a highly generalizable policy, in particular when the offline data mismatches the real environment. In the case that the pre-collected trajectories are from the expert and match the real environment, our experiments also verify that ReDM does not degrade the performance.
>
> **Q4**: Learned dynamics models may be unreasonable like violating the physics or economics laws, and *eligibility* can hardly help alleviate this issue.
>
> **A4**: We would like to clarify that the key to the success of ReDM is to avoid generating too many unreasonable dynamics models. ReDM has employed eligibility to prune some unreasonable dynamics models, which has already led to good performance in cases. If more unreasonable dynamics models can be pruned, ReDM can surely improve its efficiency. We will also explore other principles to prune more unreasonable dynamics models.
>
> We hope that our explanation and clarification have effectively addressed your concerns. We are readily available to provide further clarification or address any additional questions you may have.
>
> [1]. Rawal Khirodkar et al. Adversarial Domain Randomization.
>
> [2]. Lerrel Pinto et al. Robust Adversarial Reinforcement Learning.
>
> [3]. Marc Rigter et al. RAMBO-RL: Robust Adversarial Model-Based Offline Reinforcement Learning.
>
> [4]. Tianhe Yu et al. MOPO: Model-based Offline Policy Optimization.

---

> ### Comment · Reviewer_YzNF · 2023-11-23
>
> Thank you for the response. Most of my concerns are addressed.  Besides, you are encouraged to further explore more methods without the assumptions mentioned in my Weakness 3.  Overall, this paper is still of nice quality and I will raise my score. Good luck :)

---

### Author Response · Authors · 2023-11-19
**General response**

We sincerely thank you for your engagement in assessing our paper. In our rebuttal, we will address questions and concerns shared among all reviewers, and then provide answers to the remaining questions in respective responses. The paper has also undergone revisions accordingly to enhance the clarity of our conceptualization, including:

- **Related Work**: We have included a more detailed comparison and description of MAPLE in the main text of our paper. Additionally, we have added a discussion on unsupervised reinforcement learning in Appendix G for further elaboration.
- **Method**: We added a more detailed description of the relationship between our proposed model generation principles and the errors to be controlled. Additionally, we have supplemented formulations and descriptions in the theorems and methods as well.
- **Experiment**: We have included additional experimental details, such as specific hyperparameters, referenced performance metrics, more detailed implementation descriptions, and learning curves of ReDM-o. Moreover, we have conducted more experiments on D4RL Adroit to further demonstrate the effectiveness of our method. Additionally, we have revised the descriptions that were unclear in relation to the figures and tables.
- **Figure**: Based on the feedback from reviewer GFGi, we have incorporated an additional figure in Appendix H to offer more specific and vivid descriptions of our method.
- **Writing**: We have revised unclear statements and corrected grammar errors in the text.

With these supplementary explanations as well as experiments, we aspire to address your concern about our paper.

The primary contribution of this paper is the policy rehearsing framework that rehearses the meta-policy for various scenarios beforehand, so that the meta-policy can make adaptive decisions based on the actual outcome. We identify the most challenging part of rehearsing as the efficient generation of candidate models and propose to constrain the model generation in terms of their diversity and eligibility. For low-dimension tasks, the proposed algorithm ReDM can obtain useful policies solely by rehearsing but use zero data. In more complex tasks, we show that some forms of supervision can be seamlessly incorporated into ReDM, such as mismatching data in the ReDM-o experiments. This paper shows the possibility of achieving highly generalizable policies by rehearsing. We will explore ways to incorporate more constraints to further improve the efficiency in the future.

**Common Question 1 (Reviewer YzNF, AWzJ)**: How can ReDM learn about the target dynamics without any interaction data?

Answer: We would like to clarify that ReDM does NOT learn the target dynamics. ReDM generates diversified and eligible candidate models. It then trains a meta-policy in these models, so that the meta-policy is able to generalize over different dynamics. Without any interaction data, the generated candidate models are expected to represent the whole model space. In such cases, the meta-policy can adapt to any environment including the real environment.

**Common Question 2 (Reviewer YzNF, anZu)**: Is the assumption of known reward function, termination function, initial state distribution, and parameterized dynamics too strong?

**About reward, termination functions, and initial state distribution.** In many real-world applications of reinforcement learning, human experts are aware of the basic properties of the task, including the reward function, termination function, and initial states. These properties can often be cheaper to obtain than interaction data. Therefore, assuming known reward and termination functions are acceptable in these situations. Meanwhile, we will continue to explore more methods without these assumptions in the future.

**About parameterization of dynamics models.** We employ neural networks to represent the dynamics, which can have strong expressive power and have been extensively applied in previous model-based RL studies (e.g., MBPO, MOPO, MuZero, etc). Thus we don't think this is an overly strong assumption.

**Common Question 3 (Reviewer anZu, GFGi, AWzJ)**: A random shooting planner may not be efficient in optimizing the dynamics model for the eligibility reward in complex control tasks.

Answer: We agree that a random shooting planner can be less efficient and there are more advanced methods that can be applied instead. However, for the purpose of concept-proof and simplicity, we show that with such a simple method ReDM can also yield good results.

---

### Meta-Review · Area_Chair_uQFx · 2023-12-05

**Metareview:**

This paper proposes to find generalizable policies by rehearsing over a diverse set of hypothetical dynamics models and acting adaptively according to actual responses.  This improves the generalizability to previously unseen environments, and helps even when the available interaction data is limited or mismatched.   The experimental results corroborate the effectiveness of the approach.

This paper is well written.  The idea is neat and well motivated, improving upon MAPLE and other meta-RL methods.  Extensive experiments with both low-dimensional continuous control tasks and two D4RL tasks demonstrate the effectiveness of the method, with or without interaction data.  It is a good contribution to the area and a good addition to the conference.

**Justification For Why Not Higher Score:**

The connection and contrast with existing meta-RL methods can be better elucidated.  More D4RL experiments can be helpful.

**Justification For Why Not Lower Score:**

This paper is well written.  The idea is neat and well motivated, improving upon MAPLE and other meta-RL methods.  Extensive experiments with both low-dimensional continuous control tasks and two D4RL tasks demonstrate the effectiveness of the method, with or without interaction data.  It is a good contribution to the area and a good addition to the conference.

---

### Decision · Program_Chairs · 2024-01-16

Accept (poster)